# p21 regulates expression of ECM components and promotes pulmonary fibrosis via CDK4 and Rb

Nurit Papismadov ⓘ, Naama Levi, Lior Roitman, Amit Agrawal, Yossi Ovadya, Ulysse Cherqui ⓘ, Reut Yosef, Hagay Akiva ⓘ, Hilah Gal & Valery Krizhanovsky ⓘ ✉

## Abstract

Fibrosis and accumulation of senescent cells are common tissue changes associated with aging. Here, we show that the CDK inhibitor p21 (CDKN1A), known to regulate the cell cycle and the viability of senescent cells, also controls the expression of extracellular matrix (ECM) components in senescent and proliferating cells of the fibrotic lung, in a manner dependent on CDK4 and Rb phosphorylation. p21 knockout protects mice from the induction of lung fibrosis. Moreover, inducible p21 silencing during fibrosis development alleviates disease pathology, decreasing the inflammatory response and ECM accumulation in the lung, and reducing the amount of senescent cells. Furthermore, p21 silencing limits fibrosis progression even when introduced during disease development. These findings show that one common mechanism regulates both cell cycle progression and expression of ECM components, and suggest that targeting p21 might be a new approach for treating age-related fibrotic pathologies.

**Keywords** Cellular Senescence; p21 (CDKN1A); CDK4; Extracellular Matrix (ECM); Fibrosis
**Subject Categories** Cell Adhesion, Polarity & Cytoskeleton; Cell Cycle; Molecular Biology of Disease

See also: P Turano & U Herbig

## Introduction

Cellular senescence is a stable cell cycle arrest of cells that limits their proliferative potential (Di Micco et al, 2021; Hernandez-Segura et al, 2018). It is a complex biological process that executes both positive and detrimental effects in vivo, depending on the biological context. Senescent cells can support tissue repair by limiting the excessive proliferation of cells (Jun and Lau, 2010; Krizhanovsky et al, 2008). Conversely, the accumulation of senescent cells in tissues, a process that occurs during aging, contributes to age-related pathologies, including fibrotic diseases, and shortens healthspan and lifespan (Baker et al, 2016;

Ovadya et al, 2018). The pathogenesis of fibrotic tissue remodeling is affected by accelerated extracellular matrix (ECM) deposition in response to injury, inflammation, or disturbed homeostasis (Wynn, 2008; Wynn and Ramalingam, 2012). Fibrosis develops in response to signals from tissue-resident, circulating, or recruited cell types, including epithelial, endothelial, fibroblast and immune cell populations. During fibrotic diseases, these cells may become senescent and promote the fibrotic process (Lehmann et al, 2017; Ogrodnik et al, 2017; Schafer et al, 2017). While our knowledge of senescent cell contributions to tissue fibrosis continues to expand, the understanding of the context-specific influence of molecular mechanisms of senescence on the pathobiology of fibrosis remains a major challenge.

Cellular senescence halts cell proliferation in response to various stressors, including telomere shortening, oncogene activation and DNA damage (Burton and Krizhanovsky, 2014; Gonzalez-Gualda et al, 2021; Herranz and Gil, 2018; Munoz-Espin and Serrano, 2014; Salama et al, 2014). The senescence growth arrest is established by the p53-p21 and p16-Rb pathways and maintained by two cyclin-dependent kinase inhibitors (CDKIs) p21 (also termed CDKN1A) and p16 (also termed CDKN2A) (Childs et al, 2017; Di Micco et al, 2021; Sharpless and Sherr, 2015). These CDKIs inhibit cyclin-dependent kinases (CDKs), thus suppressing the phosphorylation of the retinoblastoma protein (Rb). This process halts cell proliferation by suppressing the activity of E2F transcription factors, which regulate cell cycle progression genes (Fischer and Muller, 2017; Otto and Sicinski, 2017; Salama et al, 2014). While the effect of CDKIs on cell cycle regulation is well established, their contribution to the regulation of the microenvironment has only started to be elucidated (Sturmlechner et al, 2021) and their effect on ECM is unknown.

One of the most devastating age-related fibrotic diseases with limited treatment options is pulmonary fibrosis (Faner et al, 2012; Hashimoto et al, 2016; Schafer et al, 2017). Pulmonary fibrosis is a progressive and ultimately fatal lung disease that results from the destruction of lung parenchyma and the accumulation of scar tissue in the lungs (Kuwano et al, 2016). Senescent cells are present in fibrotic lung tissue of human patients and in a well-defined model of pulmonary fibrosis in mice–bleomycin-induced pulmonary fibrosis (Aoshiba et al, 2003; Hecker et al, 2014; Kuwano et al, 2016). Notably, p21-positive senescent cells are present in the lung during pulmonary fibrosis (Adams et al, 2020; Kuwano et al, 2016). However, the contribution of these cells to the progression of the

Department of Molecular Cell Biology, The Weizmann Institute of Science, 7610001 Rehovot, Israel. ✉E-mail: valery.krizhanovsky@weizmann.ac.il

disease is insufficiently understood. Therefore, uncovering the potential role of p21 during pulmonary fibrosis pathology is essential for devising more effective therapies.

The contribution of p21 to fibrotic pathology might rely on its ability to regulate the multiple phenotypes of both proliferating and senescent cells. Indeed, one of the characteristics of senescent cells necessary for their persistence, resistance to cell death, which is mediated by p21 and BCL-2 family members (Chang et al, 2016; Wang, 1995; Yosef et al, 2017; Yosef et al, 2016). Of note, p21 maintains the viability of senescent cells by restricting activation of the DNA damage response (DDR) and NF-κB pathways and limits the expression of extracellular matrix (ECM) components (Yosef et al, 2017). Understanding the regulation of ECM components by p21 might be crucial in order to elucidate how the senescence program affects the extracellular microenvironment and fibrosis. To achieve this understanding, it is necessary to study the effects of p21 loss of function during the development of the disease. The consequences of time-controlled p21 silencing in vivo were not studied before, and the understanding of the possible functions of p21 in different pathologies comes from studies using p21 deficient mice. For instance, p21 knockout supports tissue regeneration and limits the aging phenotypes of Telomerase knockout mice and protects mice from the induction of liver and kidney fibrosis (Choudhury et al, 2007; Megyesi et al, 2015; Yosef et al, 2017). However, the absence of p21 prior to disease induction hampers proper evaluation of the role of p21 during the disease progression.

We aimed to understand the mechanisms that mediate the effect of p21 on the ECM component expression, lung fibrosis and inflammation in vivo. We show that p21 regulates multiple components of the extracellular microenvironment in senescent and proliferating cells via its interaction with CDK4. In fact, p21 regulates the CDK4-mediated phosphorylation of Rb that directly affects ECM component expression. To evaluate the effects of p21 in vivo, we used p21 knockout mice and an inducible p21 knockdown mouse model in conjunction with bleomycin-induced pulmonary fibrosis. Both initiation of the p21 knockdown during the disease progression and p21 knockout alleviate the lung fibrosis pathology. In both models, lack of p21 leads to a reduced accumulation of senescent cells and a decrease in the inflammatory response. Overall, these findings suggest that p21 is a key regulator of fibrosis and the cellular microenvironment.

## Results

### p21 knockout reduces the amount of senescence markers and the inflammatory responses in lung fibrosis

The contribution of p21 to the development of fibrotic pathologies might rely on its ability to regulate the resistance of senescent cells to apoptosis and limit the expression of ECM components (Yosef et al, 2017). We, therefore, aimed to determine the effect of p21 on fibrotic pathology in vivo. To this aim, we induced lung fibrosis in mice by administration of bleomycin (BLM) (1.5 u/kg), or PBS vehicle as a control, to wild-type (WT) or p21 knockout ($p21^{-/-}$) (Deng et al, 1995) 8-week-old female mice by a single intra-tracheal installation (Aoshiba et al, 2003). The mice were sacrificed after 10 days, during the inflammatory stage of the disease, or after 21 days, during the fibrotic stage of the disease (Fig. 1A). We aimed

to evaluate the extracellular microenvironment, accumulation of senescent cells, lung inflammation, and fibrosis in these mice. We observed a significant increase in p21 expression in the lung tissue at these time points, at the protein and the mRNA levels in WT BLM-treated mice compared to PBS-treated mice (Appendix Fig. S1a,b,S1c, respectively). Of note, p21 expression was significantly higher on day 10, during the inflammatory stage, than on day 21, during the fibrotic stage. Similarly, immune-fluorescent nuclear staining of p21 revealed a marked elevation in the number of p21 expressing cells at both time points, while on day 10 the expression of p21 was two fold higher than on day 21 (Appendix Fig. S1d,e). Overall, p21 expression is increased in lung fibrosis pathology following treatment with BLM.

In order to study the presence of senescent cells in the lung tissue, we evaluated the protein expression levels of the senescence markers p15, p16, and p53 and the marker of DNA damage, phospho-ATM (p-ATM), in the lungs of WT and $p21^{-/-}$ mice by immunoblot analysis (Fig. 1B; Appendix Fig. S1f), and assessed γH.2AX expression by immune-fluorescent staining (Fig. 1C,D). A significant upregulation of p15 (tenfold), p16 (tenfold) and p53 (sevenfold) protein levels was detected in BLM-treated WT mice compared to PBS-treated mice after 10 days, while p21 knockout abolished the increase in expression of these markers (Fig. 1B). This upregulation was still observed in WT mice after 21 days [p15 (sixfold), p16 (threefold), and p53 (fivefold)]. Similarly, significant p21-dependent differences were observed in mRNA expression of *p15* and *p16* (Appendix Fig. S1g). Notably, co-staining of p16 and p21 revealed a significant increase in cells expressing both markers in BLM-treated WT mice compared to PBS-treated mice (Appendix Fig. S1h,i). Furthermore, BLM treatment caused a fivefold (10 days) and threefold (21 days) increase in ATM phosphorylation (Fig. 1B; Appendix Fig. S1f), and a 40% (10 days) and 30% (21 days) increase in the amount of γH.2AX positive cells (Fig. 1D) compared with PBS-treated WT mice. The p21 knockout abolished this effect. This suggests that senescent cells accumulate during the inflammatory stage on day 10, and at the fibrotic stage, on day 21, the amount of these cells in the tissue is reduced. We then studied whether p21 affects the presence of apoptotic cells in the lung. The immunofluorescent staining of Cleaved-Caspase 3 (Cas3), a marker for apoptosis, revealed higher levels of apoptosis after 10 days in BLM-treated WT mice than after 21 days (Fig. 1C,E). There were no apoptotic cells detected in the lungs of p21 knockout mice at any of these time points, probably reflecting the lower level of tissue damage in these mice. Thus, p21 knockout reduces the presence of senescence and apoptosis markers in the lungs following BLM treatment.

One of the hallmarks of lung fibrosis is inflammation, which is characterized by the recruitment and accumulation of both innate and adaptive immune cells (Desai et al, 2018; Wynn, 2008). To examine whether p21 knockout altered the inflammatory response to BLM-induced lung injury, we analyzed the mRNA expression levels of pro-inflammatory SASP components *Ccl5*, *Cxcl5*, *Il-1α*, *Il-1β*, *Il-6* and *Kc* and *Ifn-γ* (Fig. 1F; Appendix Fig. S2a,b). We observed a significant fivefold (10 days) and threefold (21 days) increase in cytokine expression in BLM-treated WT mice compared with PBS-treated mice. Such an increase was not observed in p21 knockout mice. Of note, the number of immune foci (Fig. 1G) and of CD45+ immune cells (Appendix Fig. S2c,S2k, for 10 and 21 days, respectively) were also significantly increased in BLM-treated WT mice compared with PBS-treated mice, but not in p21

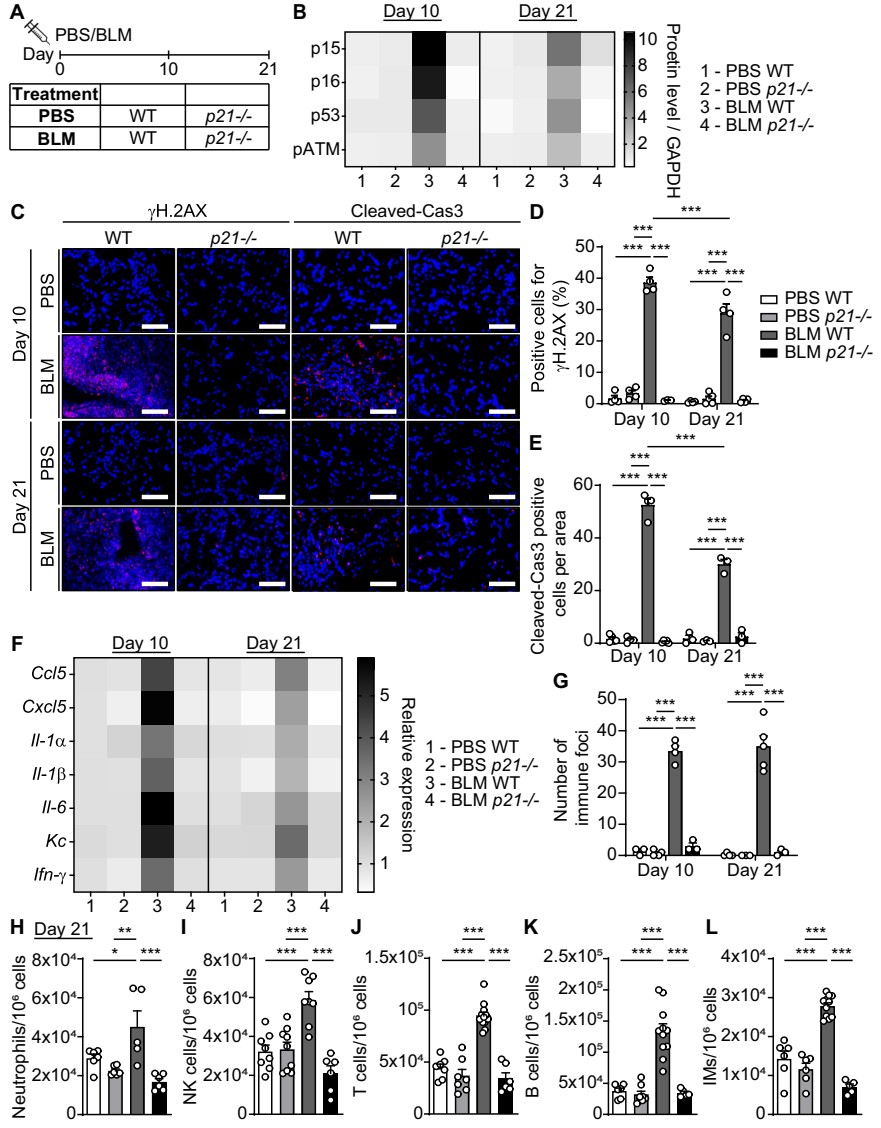

**Figure 1. p21 knockout reduces the presence of senescent markers and the inflammatory responses in bleomycin-induced lung fibrosis.**

(A) Experimental design. WT or *p21*⁻/⁻ mice were administered with bleomycin (BLM) or PBS vehicle by one intra-tracheal installation. The lungs were analyzed 10 and 21 days thereafter. (B) Quantification of protein levels of p15, p16, and p53, and phospho-ATM (p-ATM) in the mice lungs (the images are presented in Appendix Fig. S1f) (BLM WT vs. BLM *p21*⁻/⁻, p15 [$P < 0.0001$], p16 [$P < 0.0001$], p53 [$P < 0.0001$], p-ATM [$P < 0.0001$]). (C) IF staining of γH.2AX (left panel) and cleaved-caspase 3 (Cleaved-Cas3) (right panel) in the mice lungs described in (A). Scale bar: 100 μm. (D) Quantification of γH.2AX IF staining presented in (C) (BLM WT vs. BLM *p21*⁻/⁻, $P < 0.0001$). (E) Quantification of Cleaved-Cas3 IF staining presented in (C) (BLM WT vs. BLM *p21*⁻/⁻, $P < 0.0001$). (F) Relative mRNA expression levels of the indicated cytokines in the mice lungs as described in (A). (G) Number of Immune foci in lung sections of the above mice (BLM WT Day 10 vs. BLM WT Day 21, $P = 0.0001$). (H–L) The flow-cytometry analysis of cells from WT and *p21*⁻/⁻ mice lungs 21 days following BLM administration: (H) neutrophils (CD45 + /Ly6G + /CD11b + ), (I) NK cells (CD45 + /Ly6G-/Nkp46 + ), (J) T cells (CD45 + /CD3 + ), (K) B cells (CD45 + /B220 + ) and (L) interstitial macrophages (IM's) (CD45 + /CD11c + /Siglec-F-/CD11b + /CD24 + ) (BLM WT vs. BLM *p21*⁻/⁻, neutrophils [$P = 0.0003$], NK cells [$P < 0.0001$], T cells [$P < 0.0001$], B cells [$P < 0.0001$], IM's [$P < 0.0001$]). Data information: Data were analyzed using one-way ANOVA. *$P < 0.05$. **$P < 0.005$. ***$P < 0.0005$. Data are presented as mean ± SEM (B, $n = 4-6$; C-E, γH.2AX; $n = 4-5$, Cleaved-Cas3; $n = 3-5$; F, $n = 3-6$; G, $n = 3-5$; H-L, $n = 5-12$ independent repeats). Source data are available online for this figure.

knockout mice. We then analyzed the presence of specific components of the immune system in the lungs by flow cytometry. This analysis revealed an accumulation of neutrophils, NK cells, CD3 + , CD4 + , and CD8 + T cells, B cells, and interstitial macrophages (IM's) (Appendix Fig. S2d–j [10 days], Fig. 1H–L, Appendix Fig. S2k–m [21 days]) following BLM administration. Remarkably, p21 knockout resulted in a significantly lower accumulation of all these immune cell subsets. Therefore, p21

knockout alleviates the accumulation of immune cells and inflammation in the lungs following BLM administration.

## p21 knockout limits the development of bleomycin-induced lung fibrosis

To understand the effect of p21 knockout on the development of lung fibrosis, we monitored the body weight of the mice, an

indicator of the severity of lung impairment by the BLM challenge (Sikic et al, 1978). As expected, BLM-treated WT mice lost an average of 6 g, 21 days post BLM injection. Notably, p21 knockout mice showed an initial weight loss, but gained their weight back starting from day 14, until reaching their original weight by day 21 (Fig. 2A). To evaluate the fibrosis in these mice we stained lung sections by hematoxylin and eosin (H&E) and by Sirius Red, which stains the ECM. As expected, a disruption of the lung architecture and an accumulation of fibrotic tissue were observed in WT mice 21 days following BLM administration (Fig. 2B,C). An accumulation of fibrotic tissue was also observed 10 days following BLM administration but was significantly lower than after 21 days (Fig. 2B,C). Strikingly, the Sirius Red-stained area was significantly reduced in the lungs of BLM-treated $p21^{-/-}$ mice compared to WT mice (Fig. 2C) and the lung architecture of these mice was preserved (Fig. 2B). Therefore, the lack of p21 limits the development of fibrosis in the lung. The reduction in the Sirius Red staining was accompanied by a significant decrease in mRNA expression of molecular markers of fibrosis, including ECM components (Fig. 2D; Appendix Fig. S3a,b) and fibroblast marker *Pdgf-rα* (Fig. 2E) at both days 10 and 21 post BLM. The p21 knockout also led to a fourfold and sixfold decrease in protein expression of COL1 and Alpha smooth muscle actin (αSMA, marks activated fibroblasts), respectively, (Fig. 2F; Appendix Fig. S3c), and to a significant decrease in immune-fluorescent staining of αSMA (Fig. 2G,H). Therefore, p21 knockout alleviates BLM-induced inflammation, accumulation of senescent cells, and lung fibrosis.

We then set to understand which cells in the lung might be affected by the p21 knockout and contribute to fibrosis. Analysis of the single-cell RNA-seq dataset, published in Strunz et al (Strunz et al, 2020) revealed that ECM components are expressed in the same cell populations as p21 (Appendix Fig. S4a–d). This analysis also highlighted the increase in ECM components expression in stromal cells (fibroblasts) and alveolar macrophages, comparing to the other cell populations in the lung, as well as the increase in the number of these cells during fibrosis development (Appendix Figs. S4e–g and S5). Therefore, both macrophages and fibroblasts express p21 and contribute to the development of fibrosis in this model.

## p21 regulates ECM components expression in both senescent and proliferating cells via CDK4

p21 regulates the cell cycle by inhibition of CDKs function. However, p21 also maintains the viability of DNA damage-induced senescent (DIS) cells, regulates the expression of genes encoding ECM components (Yosef et al, 2017), and promotes the lung fibrosis pathology. We therefore set to understand the molecular mechanisms behind the effects of p21 on the extracellular microenvironment of senescent cells. To do so, we first aimed to identify the binding partners of p21 via a pull-down assay coupled with mass spectrometry (MS) based protein identification. We extracted proteins from DIS human primary BJ fibroblasts treated with the DNA-damaging agent etoposide, and from proliferating (G) control cells, and precipitated these proteins using an anti-p21 antibody. We observed an increase in p21 in senescent cells both in total extracts and following the immunoprecipitation (Fig. 3A). We then identified the p21-bound proteins in senescent cells by MS. This analysis identified that the strongest p21 interacting partners

in senescent cells were Cyclin-Dependent Kinase 4 (CDK4) and CyclinD1 (CCND1) (Fig. 3B). Surprisingly, CDK2, another CDK regulated by p21, was not identified in this assay. We therefore hypothesized that regulation of the extracellular microenvironment by p21 might be mediated by its direct interaction with CDK4. To test this hypothesis, we treated DIS BJ primary human fibroblast cells with siRNA mixes targeting p21 (sip21), CDK4 (siCDK4) CyclinD1 (siCCND1) alone or in combinations, which achieved an efficient knockdown, or control siRNAs (siCtrl) (Appendix Fig. S6a). As expected, p21 knockdown resulted in senescent cells' death, unlike in control cells (Appendix Fig. S6b). Notably, the combined knockdown of p21 and CDK4, p21 and CCND1, or a triple knockdown, all resulted in complete rescue of the senescent cells from p21 knockdown-induced death. To test the involvement of CDK4 in the regulation of ECM components, we measured the protein levels of collagen-1 (COL1) in DIS cells following p21 and CDK4 knockdown. Remarkably, knockdown of p21 caused a significant, 66% reduction in COL1 protein expression, while the combined knockdown of p21 and CDK4 or the knockdown of CDK4 alone had no such effect (Fig. 3C,D; Appendix Fig. S6c). We then measured the mRNA expression levels of ECM components *COL1A1*, *COL3A1*, *COL4A1*, and *FN1* in this experimental system. We observed that p21 regulates the mRNA expression of these ECM components in a CDK4-dependent manner, similarly to what we observed with COL1 protein level (Fig. 3E). To verify that the CDK4-CCND1 complex mediates the observed effect we knocked down p21 together with CCND1 and observed that the CCND1 knockdown mimicked the effect of CDK4 knockdown on the expression of ECM components (Appendix Fig. S6d).

To determine whether p21 regulation of the ECM expression in senescent cells is restricted to BJ cells we also analyzed the expression of ECM components in DIS human IMR-90 fibroblasts treated with sip21 and siCDK4 mixes (Appendix Fig. S6e). We observed that, similarly to BJ cells, p21 regulates ECM component expression in a CDK4-dependent manner at both the mRNA and protein levels (Fig. 3F; Appendix Fig. S6f–h, respectively). Finally, to find out whether the regulation of the ECM components is dependent on the time of p21 knockdown, we analyzed cells that were already lacking p21 prior to DNA damage induction. We treated mouse lung fibroblast cells (MLFs) derived from wild-type (WT) and p21 knockout ($p21^{-/-}$) mice with etoposide and then with CDK4/6 inhibitor Abemaciclib (CDKi) upon senescence induction (Appendix Fig. S6i). CDK4/6 inhibition in $p21^{-/-}$ cells resulted in a significant increase in the mRNA expression levels of the ECM components (Fig. 3G) and in COL1 protein levels (Appendix Fig. S6j–l). Together, these results indicate that p21 regulates the expression of ECM-encoding genes in senescent cells in a CDK4-dependent manner.

To understand if the effect of p21 on the extracellular microenvironment is limited to senescence, we transduced proliferating cells with a siRNA mix against p21 or a control mix and measured the expression of ECM components by RT-PCR analysis (Appendix Fig. S7a). A significant decrease in the expression of ECM components, 46% on average, was detectable already 24 h following p21 knockdown (Appendix Fig. S7b). We then set to determine whether the effect of p21 on the ECM components in proliferating cells is mediated by CDK4. We transduced proliferating cells with sip21 and siCDK4 siRNA mixes. The efficiency of the knockdown was confirmed by analyzing the

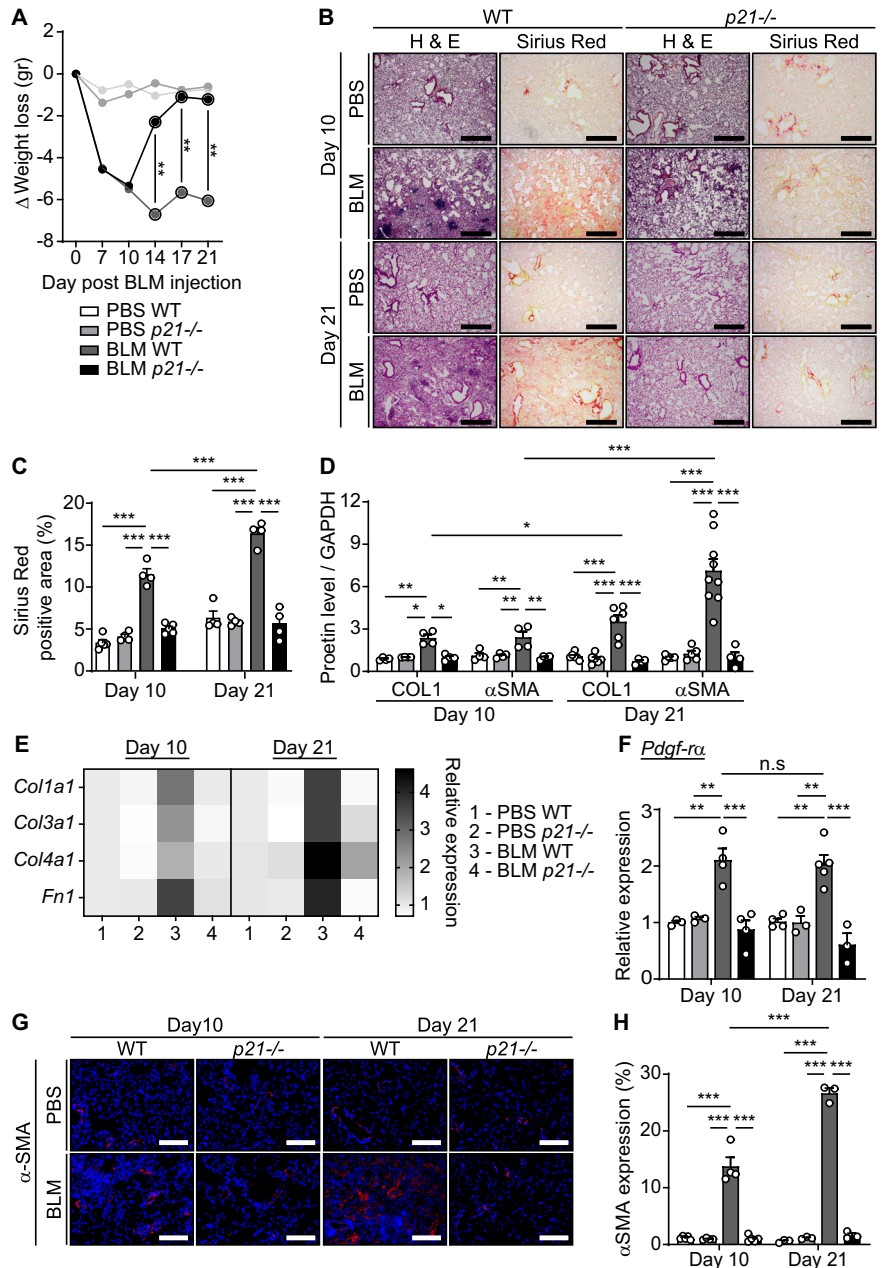

**Figure 2. p21 knockout protects lungs from bleomycin-induced fibrosis.**

WT or $p21^{-/-}$ mice were administered with bleomycin (BLM) or PBS vehicle by one intra-tracheal installation. (**A**) Body weight was monitored every 4 days and is presented as delta (Δ) from the starting weight (BLM WT vs. BLM $p21^{-/-}$, day 14 [$P = 0.006$], day 17 [$P = 0.004$], day 21 [$P = 0.008$]). The lungs were analyzed at days 10 and 21. (**B**) Lung sections from WT and $p21^{-/-}$ mice treated with BLM or PBS were stained with H&E and Sirius Red. Scale bar: 200 μm. (**C**) Quantification of Sirius Red staining presented in (**B**) (BLM WT Day 10 vs. BLM WT Day 21, $P < 0.0001$). (**D**) Quantification of protein levels of collagen-1 (COL1), and α-SMA at day 10 and day 21 following the BLM administration (BLM WT Day 10 vs. BLM WT Day 21, COL1 [$P = 0.036$], α-SMA [$P < 0.0001$]). (**E, F**) mRNA expression levels of (**E**) collagen-1 (*Col1a1*), collagen-3 (*Col3a1*), collagen-4 (*Col4a1*) and Fibronectin-1 (*Fn1*) and of (**F**) *Pdgf-rα* in the mice lungs (BLM WT vs. BLM $p21^{-/-}$, day 10 [$P = 0.0001$], day 21 [$P = 0.0001$]). (**G**) IF staining of lung sections for α-SMA. Scale bar: 200 μm. (**H**) Quantification of α-SMA expression as presented in (**G**) (BLM WT Day 10 vs. BLM WT Day 21, $P < 0.0001$). Data information: Data were analyzed using one-way ANOVA. *$P < 0.05$. **$P < 0.005$. ***$P < 0.0005$. Data are presented as mean ± SEM (**A**, $n = 4$–14; **B**, **C**, $n = 4$–5; **D**, $n = 3$–8; **E**, $n = 3$–5; **F**, $n = 4$–9; **G**, **H**, $n = 3$–5 independent repeats). Source data are available online for this figure.

mRNA expression levels (Appendix Fig. S7c). Similarly to the effect observed in senescent cells, a significant reduction in COL1 protein levels (Fig. 3H,I; Appendix Fig. S7d) and in the mRNA expression levels of ECM components *COL1A1*, *COL3A1*, *COL4A1*, and *FN1*

(Fig. 3J) was detected following p21 knockdown, while the combined knockdown of p21 and CDK4 abolished this effect. Thus, the CDK4-mediated effect of p21 knockdown on the ECM components expression is not limited to senescent cells.

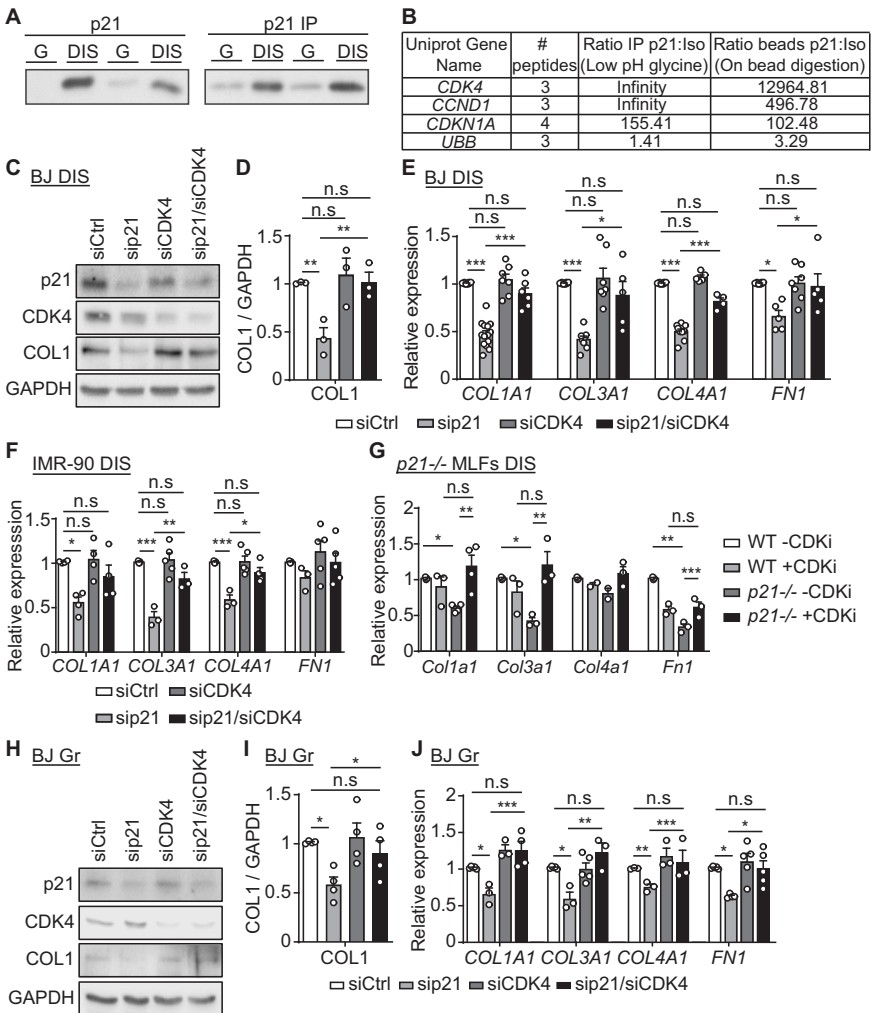

**Figure 3. p21 regulates the ECM components expression in both proliferating and senescent cells via CDK4.**

(A) Protein expression and immunoprecipitation (IP) of p21 from DIS and control (G) BJ cells. p21 expression levels were tested by western blot (left panel). Immunoprecipitation was performed using anti-p21 antibody and the resulted precipitate was analyzed by western blot (right panel). (B) Proteomic analysis identified p21-bound proteins. The numbers are ratios of reads between proteins bound to anti-p21 antibody vs. isotype control using two indicated methods of elution. (C) DIS BJ cells were transduced with siRNAs targeting p21 (sip21), CDK4 (siCDK4), or control siRNAs (siCtrl) and protein levels of p21, CDK4 and collagen-1 (COL1) were analyzed by western blots. (D) Quantification of protein levels of COL1 relative to GAPDH control (sip21 vs. sip21/siCDK4, $P = 0.004$). (E) mRNA expression levels of collagen-1 (COL1A1), collagen-3 (COL3A1), collagen-4 (COL4A1) and Fibronectin-1 (FN1) relative to the control in DIS BJ cells as described in (C) (sip21 vs. sip21/siCDK4, COL1A1 [$P < 0.0001$], COL3A1 [$P = 0.013$], COL4A1 [$P < 0.0001$], FN1 [$P = 0.046$]). (F) mRNA expression levels of the indicated ECM components in DIS IMR-90 fibroblasts following treatment with siRNA targeting p21, CDK4, or their combinations relative to the control (sip21 vs. sip21/siCDK4, COL3A1 [$P = 0.004$], COL4A1 [$P = 0.01$]). (G) mRNA expression levels of the indicated ECM components relative to control in mouse lung fibroblast cells (MLFs) derived from WT or $p21^{-/-}$ mice incubated with or without Abemaciclib ($+/-$CDKi) ($p21^{-/-}$ -CDKi vs. $p21^{-/-}$ +CDKi, Col1a1 [$P = 0.06$], Col3a1 [$P = 0.005$], Fn1 [$P = 0.009$]). (H) Western blot analysis of p21, CDK4 and COL1 in proliferating (Gr) BJ cells treated with siRNAs targeting p21, CDK4, or their combination. (I) Quantification of protein levels of COL1 relative to GAPDH control (sip21 vs. sip21/siCDK4, $P = 0.044$). (J) mRNA expression levels of the indicated ECM components relative to the control in proliferating (Gr) BJ cells as described in (H) (sip21 vs. sip21/siCDK4, COL1A1 [$P = 0.002$], COL3A1 [$P = 0.001$], COL4A1 [$P = 0.001$], FN1 [$P = 0.028$]). Data information: Data were analyzed using one-way ANOVA. *$P < 0.05$. **$P < 0.005$. ***$P < 0.0005$. Data are presented as mean ± SEM (A, B, $n = 3$; C, D, $n = 3$; E, $n = 4$–14; F, $n = 3$–5; G, $n = 3$–4; H, I, $n = 4$; J, $n = 3$–5 independent repeats). Source data are available online for this figure.

## Rb is essential for the p21-mediated regulation of ECM components expression

In order to regulate the cell cycle, p21 interacts with CDK4, resulting in the inhibition of CDK-dependent phosphorylation of Rb, allowing binding of Rb to E2F and other transcription factors (Burton and Krizhanovsky, 2014; Otto and Sicinski, 2017; Sanidas et al, 2019). To test whether Rb phosphorylation plays a role in the p21-mediated changes in the extracellular microenvironment, we first knocked down CDK4 individually or together with p21 and analyzed Rb phosphorylation. Knockdown of p21 in DIS cells resulted in an increase in phosphorylated Rb protein levels, but the combined knockdown of p21 and CDK4 abolished this effect (Fig. 4A). We then set to test the role of Rb in the regulation of ECM components expression downstream to p21. We knocked down Rb (siRb) individually or together with p21 and confirmed

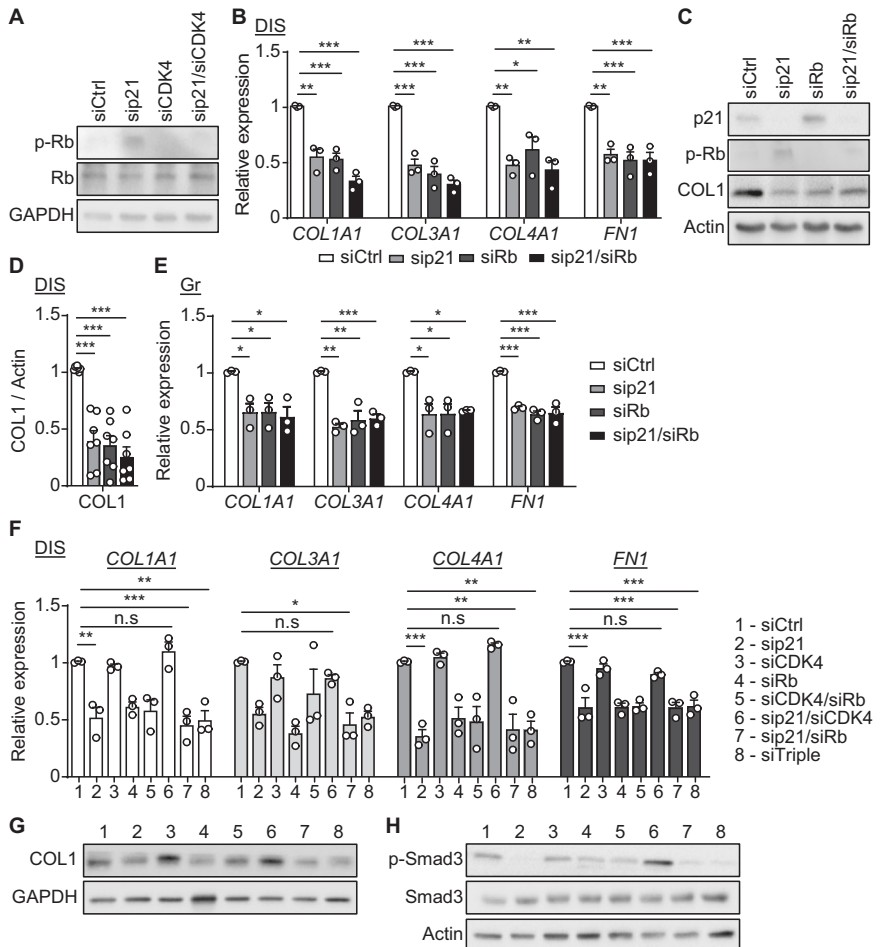

**Figure 4. Rb phosphorylation downstream to p21 regulates the ECM components expression in proliferating and senescent cells.**

(A) DIS BJ cells were transduced with siRNAs targeting p21 (sip21), CDK4 (siCDK4), or control siRNAs (siCtrl). Protein levels of phospho-Rb (p-Rb) and Rb following treatment with siRNA targeting p21, CDK4, or their combination were analyzed by western blots. (B) mRNA expression levels of collagen-1 (COL1A1), collagen-3 (COL3A1), collagen-4 (COL4A1) and Fibronectin-1 (FN1) in DIS BJ cells that were transduced with siRNAs targeting p21, Rb (siRb), or their combination relative to the control siRNAs (siCtrl vs. sip21/siRb, COL1A1 [P < 0.0001], COL3A1 [P < 0.0001], COL4A1 [P = 0.003], FN1 [P = 0.001]). (C) Protein levels of p21, p-Rb and collagen-1 (COL1) of DIS BJ described in (B). (D) Quantification of protein levels of COL1 relative to Actin control (siCtrl vs. sip21/siRb, P = 0.0001). (E) mRNA expression levels of the indicated ECM components in proliferating (Gr) BJ cells treated with siRNAs targeting p21, Rb, or their combination relative to the control (siCtrl vs. sip21/siRb, COL1A1 [P = 0.019], COL3A1 [P = 0.0013, COL4A1 [P = 0.02], FN1 [P = 0.0001]). (F) mRNA expression levels of the indicated ECM components in DIS BJ cells following treatment with siRNA targeting p21, CDK4, Rb, or their combinations (siCtrl vs. siTriple, COL1A1 [P = 0.0019], COL4A1 [P = 0.002], FN1 [P = 0.0003]). (G) Western blot analysis of COL1 in DIS BJ cells as described in (F). (H) Western blot analysis of phospho-Smad-3 (p-Smad-3) and Smad-3 in DIS BJ cells as described in (F). Data information: Data were analyzed using one-way ANOVA. *P < 0.05. **P < 0.005. ***P < 0.0005. Data are presented as mean ± SEM (A, n = 6; B, n = 3; C, D, n = 7; E, n = 3; F, n = 3; G, n = 5; H, n = 3 independent repeats). Source data are available online for this figure.

the efficiency of the knockdown (Appendix Fig. S8a,b). As expected, p21 knockdown led to a significant reduction in the mRNA expression levels of ECM components in DIS cells. However, the combined knockdown of p21 and Rb did not reverse this effect. Moreover, Rb knockdown alone caused a significant reduction in the expression of ECM components, similarly to the p21 knockdown (Fig. 4B). The analysis of the expression of COL1 protein showed a similar result (Fig. 4C,D). These results suggest that in DIS cells Rb promotes ECM components expression, downstream to CDK4 and p21. To understand whether the Rb-mediated regulation of ECM components' expression is limited to DIS cells, we performed the same set of experiments in proliferating BJ human fibroblasts (Appendix Fig. S8c). All three treatments resulted in a significant downregulation of ECM

components, similar to the effect observed in DIS cells (Fig. 4E). Therefore, our results show that Rb regulates the ECM components expression in both proliferating and senescent cells downstream to p21.

In order to identify the mechanism behind this regulation of ECM components expression, we treated DIS cells with siRNA mixes targeting p21, CDK4 and Rb alone or in combinations, or control siRNAs. The efficiency of the knockdown was confirmed by analysis of the mRNA and protein expression levels (Appendix Fig. S8d–f). Notably, Rb knockdown resulted in senescent cells' death, similar to p21 knockdown (Appendix Fig. S8g). In addition, while a combined knockdown of p21 and CDK4 resulted in a significant upregulation in mRNA expression of ECM components comparing to p21 knockdown alone, the combined triple

knockdown of p21, CDK4 and Rb reversed this effect and resulted in a significant decrease in expression comparing to the siCtrl (Fig. 4F). The analysis of the expression of COL1 protein showed a similar result (Fig. 4G; Appendix Fig. S8h). Therefore, p21 knockdown leads to Rb phosphorylation which directly affects ECM components expression.

One of the main regulators of ECM components expression is the TGF-β pathway (Ikushima and Miyazono, 2010). The TGF-β receptor regulates the expression of ECM components through phosphorylation at the C-terminus (Serine 423/425) of the transcription factor Smad-3 upon ligand binding (Massague et al, 2005). Of note, hypo-phosphorylated Rb can bind to Smad-3 and regulate its transcriptional activity (Sturmlechner et al, 2021). p21 knockdown in senescent cells leads to the downregulation of the TGF-β pathway (Yosef et al, 2017). To understand whether the p21-mediated regulation of the ECM components expression is mediated by the interplay between Rb and Smad-3, we analyzed the phosphorylation of Smad-3 following knockdown of p21, CDK4 and Rb, alone or in combination. Remarkably, the reduction in COL1 protein was accompanied with a decrease in the phosphorylation of Smad-3 (Fig. 4H; Appendix Fig. S8i) following p21 knockdown, Rb knockdown and the combined triple knockdown of p21, CDK4 and Rb. Overall, this suggests that p21 regulates the ECM components expression in a CDK4- and Rb-dependent manner.

## p21 knockdown reduces the number of senescent cells in bleomycin-induced lung fibrosis

p21 plays a pivotal role in ECM components expression in both proliferating and senescent cells. Furthermore, p21 knockout limits the induction of lung fibrosis. However, the absence of p21 prior to disease induction hampers proper evaluation of the role of p21 during later stages of disease. Therefore, we set to understand the effect of p21 knockdown during the development of lung fibrosis. To do this, we developed an inducible short hairpin RNA (shRNA) mediated p21 knockdown mice. The transgene includes a targeting construct for the best possible miR30-based shRNA targeting p21, linked to a GFP fluorescent reporter, all under a Tetracycline Response Element (TRE) promoter (Fig. 5A) (Dow et al, 2012; Premsrirut et al, 2011). We crossed these mice with the CAG promoter-driven reverse tetracycline-controlled Trans-activator 3 (rtTA3) allele, expressed in all the tissues of the mouse (Premsrirut et al, 2011).

In order to evaluate the functionality of this system, we first treated 8-week-old double-transgenic *shp21/rtTA3* and transgenic *shp21* female and male mice with Doxycycline (Dox) for a period of 3 days. This experiment showed that *shp21* was expressed in the lungs of *shp21/rtTA3* mice, but not in *shp21* mice following Dox administration (Appendix Fig. S9a). We then assessed whether the knockdown of p21 can be achieved following the activation of DDR. For that, we administered Dox to 8-week-old *shp21/rtTA3* female mice, for 4 days. The mice were irradiated (8 Gy) on day 3 and the lungs were analyzed 16 h following the irradiation (Appendix Fig. S9b). Following Dox administration, *shp21/rtTA3* mice expressed GFP (Appendix Fig. S9c,d) and *shp21* (Appendix Fig. S9e). The expression of *shp21* led to a significant decrease in *p21* mRNA (Appendix Fig. S9f) and protein levels (Appendix Fig. S9g,h) in the lungs. Therefore, this model of p21

downregulation in vivo was functional and we were able to induce knockdown of p21 in damaged lungs.

We then used this model to test the effects of p21 knockdown on the development of lung fibrosis. To this aim BLM (1.5 u/kg) or PBS vehicle as a control, were administered to 8-week-old female mice, separated into six different genotypes/treatments which were assigned into three groups (Fig. 5B): (i) PBS control (also PBS C): double-transgenic *shp21/rtTA3* mice administered with PBS and either treated or not treated with Dox; (ii) p21 control (also p21 C): single-transgenic *shp21* mice administered with BLM either treated or not treated with Dox, and double-transgenic *shp21/rtTA3* mice administered with BLM and not treated with Dox; (iii) p21 knockdown (also p21 KD): double-transgenic *shp21/rtTA3* mice administered with BLM and treated with Dox. In order to determine the best time point to initiate p21 knockdown during the development of the disease, we analyzed the expression of p21 using a single-cell RNA-seq dataset, published by Strunz et al (Strunz et al, 2020). This analysis revealed that while some increase in p21 expression was detected at day 3 following BLM administration to mice (Appendix Fig. S10a), a profound increase in p21 expression was observed in epithelial cells, endothelial cells, stromal cells, type II pneumocytes and in immune cells on day 10 following BLM administration (Appendix Fig. S10b), compared to day 21 (Appendix Fig. S10c). Indeed, immune-fluorescent nuclear staining of p21 (Appendix Fig. S10d,e) and mRNA levels of *p21* (Appendix Fig. S10f) revealed a marked elevation in p21 expression on day 10. Therefore, mice were treated with Dox starting from 10 days post BLM administration and were analyzed on day 21 (Fig. 5B).

BLM administration caused a significant increase in p21 expression in mice from p21 C group comparing to the PBS C group, which was significantly decreased in mice from the p21 KD group comparing to p21 C group (Fig. 5C–E). The expression of p21 was not significantly different in p21 KD mice comparing to PBS C mice. To evaluate the accumulation of p21 expressing cells following BLM administration, we performed an immune-fluorescent nuclear staining of p21 in all three experimental groups of mice (Fig. 5F [left panel]; Appendix Fig. S11e,f). Quantification of the staining revealed a significant decrease in the number of p21 expressing cells in the p21 KD group compared to the p21 C group (Fig. 5G). Therefore, p21 knockdown represses the accumulation of p21-positive cells in lung fibrosis following treatment with BLM.

Senescent cells exacerbate lung fibrosis pathology, and their elimination limits the disease progression (Schafer et al, 2017). Furthermore, p21 knockdown in vitro results in the death of senescent cells (Yosef et al, 2017). Therefore, we evaluated whether p21 knockdown following BLM administration leads to a reduction in the amount of senescent cells in the lung tissue. BLM treatment led to a significant elevation in p15, p16 and γH.2AX expressing cells in mice from p21 C group, compared to PBS C group (p15: Appendix Fig. S11a,c; p16: Fig. 5F [middle panel], 5H; γH.2AX: Fig. 5F [right panel], 5I). Expression of these markers was detected in cells in both alveolar (Appendix Fig. S11e) and bronchial (Appendix Fig. S11f) compartments. Remarkably, p21 knockdown abolished this effect, and resulted in a significant reduction in the expression of all these markers. This reduction was accompanied by a significant decrease in the mRNA expression of *p15* and *p16* (Fig. 5J) and in the protein levels of p15, p16, p19 (Fig. 5K,L), p-ATM and γH.2AX (Appendix Fig. S11g,h) in the lung tissue of

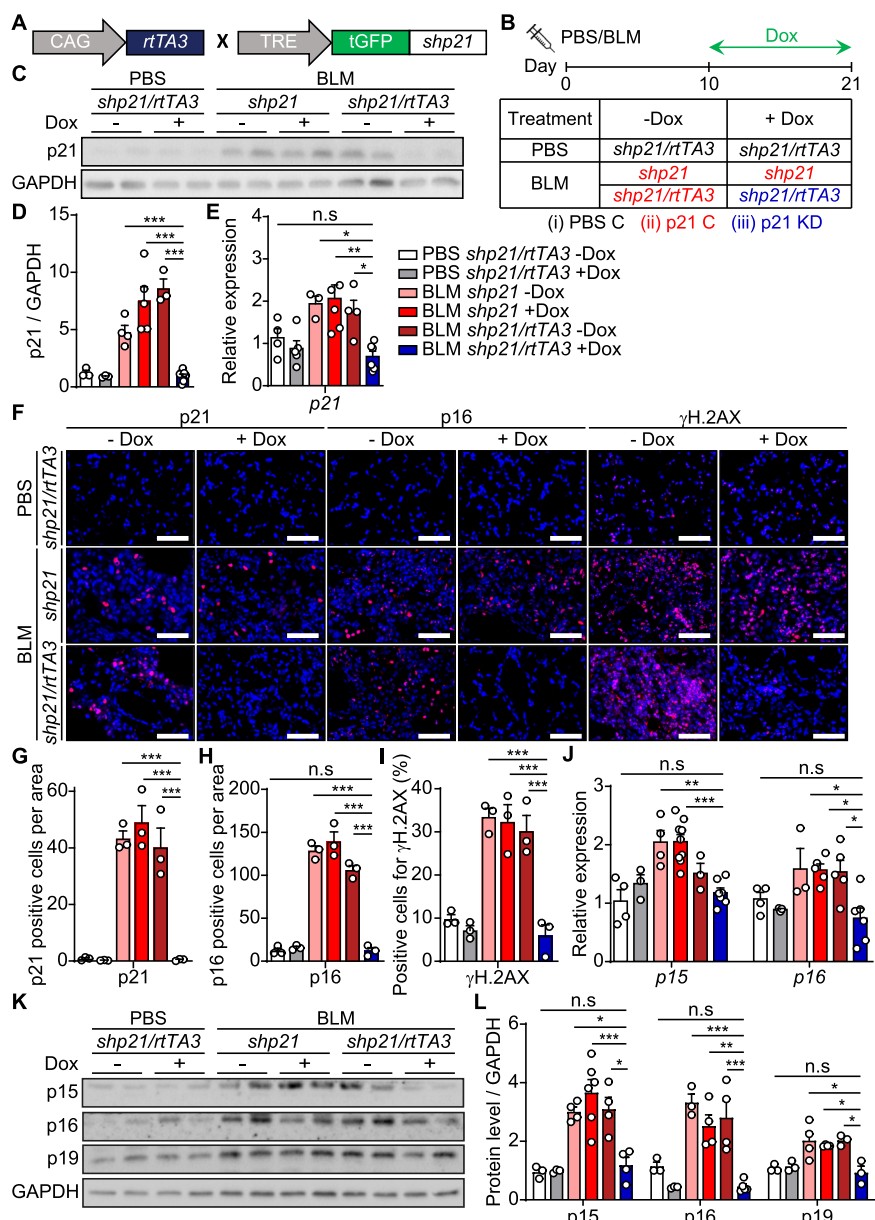

**Figure 5. p21 knockdown leads to a reduction in the presence of senescent markers in bleomycin-induced lung fibrosis.**

(A) Breeding strategy: mice carrying the *TRE-shp21* gene were crossed with mice carrying a *CAG-rtTA3* gene to generate double-transgenic mice. (B) Experimental design. Mice were administered once with bleomycin (BLM) or PBS vehicle and 10 days later Doxycycline (Dox) treatment in drinking water was initiated and continued for 11 days. Lungs were analyzed 21 days post BLM administration. The table presents the experimental groups by genotype and the type of treatments. (C) Western blot analysis of p21 in the mice lungs described in (B). (D) Quantification of protein levels of p21 relative to GAPDH control (BLM *shp21/rtTA3* –Dox vs. BLM *shp21/rtTA3* +Dox, $P < 0.0001$) and (E) mRNA expression levels of *p21* relative to PBS controls in the mice lungs described in (B) (BLM *shp21/rtTA3* –Dox vs. BLM *shp21/rtTA3* +Dox, $P = 0.046$). (F) IF staining of lung sections for p21 (left panel), p16 (middle panel) and γH.2AX (right panel). Scale bar: 100 μm. (G–I) Quantification of p21 (G), p16 (H), and γH.2AX (I) IF staining presented in (F) (BLM *shp21/rtTA3* –Dox vs. BLM *shp21/rtTA3* +Dox, p21 [$P = 0.0001$], p16 [$P < 0.0001$], γH.2AX [$P = 0.0003$]). (J) mRNA expression levels of *p15* and *p16* relative to PBS controls in the mice lungs described in (B) (BLM *shp21* +Dox vs. BLM *shp21/rtTA3* +Dox, *p15* [$P = 0.0003$], *p16* [$P = 0.017$]). (K) Western blot analysis of p15, p16, and p19 in the mice lungs described in (B). (L) Quantification of protein levels of p15, p16, and p19 relative to GAPDH control (BLM *shp21/rtTA3* –Dox vs. BLM *shp21/rtTA3* +Dox, p15 [$P = 0.015$], p16 [$P = 0.0005$], p19 [$P = 0.023$]). Data information: Data were analyzed using one-way ANOVA. *$P < 0.05$. **$P < 0.005$. ***$P < 0.0005$. Data are presented as mean ± SEM (C, D, $n = 3$–8; E, $n = 3$–6; F–I, $n = 3$; J, $n = 3$–9; K, L, $n = 3$–6 independent repeats). Source data are available online for this figure.

the p21 KD mice group comparing to the p21 C mice group. To extend the evaluation of senescence markers we stained the lungs for SA-β-gal activity. BLM treatment led to an increase in the SA-β-gal activity in the lung tissue, while a significant decrease was observed in the p21 KD mice group (Appendix Fig. S11b,d). Thus, p21 knockdown in vivo leads to a reduction in the amount of senescent cells in the lungs following BLM administration.

## p21 knockdown reduces inflammatory responses in the fibrotic lungs

To examine whether p21 knockdown altered the inflammatory response of BLM-induced lung injury, we assessed the infiltration of both innate and adaptive immune cells to the lung tissue and the expression of inflammation-associated cytokines and chemokines. We observed a significant decrease in the mRNA expression levels of the pro-inflammatory molecules *Ccl5*, *Cxcl5*, *Il-1α*, *Il-1β*, *Il-6*, and *Kc* (Fig. 6A), and a significant increase in the inflammatory-modulating molecules *Il-10*, *Cxcl9*, and *Cxcl10* (Fig. 6B) in the p21 KD mice comparing to the p21 C mice. Accumulation of immune cells in a tissue can attest the inflammation level in this tissue. Of note, the numbers of immune foci (Fig. 6C) and CD45+ immune cells (Fig. 6D) were significantly increased in mice in group p21 C, while in the p21 KD mice, they were not different from PBS C mice. We then analyzed the presence of specific components of the immune system in the lungs by flow cytometry. This analysis revealed an accumulation of B cells (Fig. 6E), CD3 + , CD4 + , and CD8 + T cells (Fig. 6F–H), neutrophils (Fig. 6I), NK cells (Fig. 6J) and interstitial macrophages (IM's) (Fig. 6K) following BLM administration. Remarkably, p21 knockdown resulted in a significantly lower accumulation of all these immune cell subsets (Fig. 6E–K). Therefore, p21 knockdown results in an attenuated immune response in the lungs following administration of BLM.

## p21 knockdown alleviates bleomycin-induced lung fibrosis

In order to understand the effect of the p21 knockdown on the progression of lung fibrosis following BLM administration, we evaluated disease parameters in every group of mice. Monitoring of body weight, revealed that mice in group p21 C lost an average of 5.2 g 21 days post BLM injection. Notably, mice in group p21 KD did not only start to gain weight following Dox administration but gained significantly more weight compared to p21 C mice by day 21 (Fig. 7A). To examine the role of p21 in the development of lung fibrosis pathology, lung sections were analyzed by H&E and Sirius Red staining. This analysis showed an alteration in lung structure, including irregular thickening of tissue between alveoli, damage to bronchi structure and large areas of fibrotic scaring, identified by the Sirius Red, in mice from group p21 C (Fig. 7B). The p21 knockdown led to a significant reduction in Sirius Red staining (Fig. 7B,C) and lung architecture of the p21 KD mice was similar to the mice from the PBS C group. Fibrosis severity was also evaluated on H&E lung sections according to the Ashcroft score (Ashcroft et al, 1988), which utilizes a numerical scale from 0 to 8 to grade fibrosis. p21 knockdown led to a significantly lower Ashcroft score compared to mice from group p21 C (Fig. 7D). The reduction in Sirius Red staining and in the Ashcroft score were accompanied by a significant decrease in mRNA expression

of molecular markers of fibrosis in the lungs of p21 KD mice compared to p21 C mice (ECM components, Fig. 7E; *Pdgf-rα*, Fig. 7F). We then evaluated the expression of the pro-fibrotic cytokine *Tgf-β* and the anti-fibrotic cytokine *Tnf-α*. The expression of *Tgf-β* was significantly reduced and the expression of *Tnf-α* was significantly increased in the p21 KD mice compared to the p21 C mice (Fig. 7G). Moreover, we assessed the changes in protein levels of COL1 and α-SMA, the marker of activated fibroblasts, by immunoblot analysis and observed a significant fivefold reduction in their expression in the p21 KD mice comparing to the p21 C mice (Fig. 7H,I). To reveal the presence of activated fibroblasts following BLM administration, we also analyzed lung sections by immunostaining for α-SMA. The lungs of the PBS C mice group were almost lacking the α-SMA staining while the lungs of the p21 C mice group showed a significant increase of the staining (Fig. 7J,K). Strikingly, p21 knockdown reverted this effect, and no significant change in α-SMA staining was observed between the p21 KD and PBS C mice groups. Altogether, it suggests that p21 knockdown induced during the development of BLM-induced lung fibrosis, limits inflammation, ECM accumulation, disruption of tissue architecture, and overall disease progression.

# Discussion

Fibrosis is the most common tissue manifestation of organismal aging. The long-term persistence of senescent cells is yet another manifestation of aging, which promotes the development of age-related pathologies, including fibrotic diseases (Calcinotto et al, 2019; Munoz-Espin and Serrano, 2014; Ovadya and Krizhanovsky, 2014). Here, we show that the CDK inhibitor p21, which maintains senescent cells viability (Yosef et al, 2017), also regulates the expression of ECM components in a CDK4- and Rb-dependent manner. We observed that p21 knockdown leads to Rb phosphorylation which directly impacts ECM components expression by altering Smad-3 phosphorylation. Furthermore, an inducible p21 knockdown mouse model revealed that p21 knockdown during the development of BLM-induced lung fibrosis reduces the expression of senescence markers and ECM components, limits inflammation, and alleviates the fibrosis pathology. Overall, these findings suggest that p21 is a key regulator of the senescent cell microenvironment and a major driver of age-related fibrotic diseases.

The most well-studied function of p21 is the regulation of CDK/Cyclin complexes during cell proliferation (Gonzalez-Gualda et al, 2021; Hernandez-Segura et al, 2018; Munoz-Espin and Serrano, 2014). CDK4 and CDK2 are members of such complexes that can phosphorylate Rb and function during the G1 and S phases of the cell cycle (Dyson, 2016; Otto and Sicinski, 2017). p21 can negatively regulate the cell cycle progression by inhibiting the activity of CDK2/CyclinE complexes (Harper et al, 1993). We show that p21 affects ECM components expression via CDK4/CyclinD1 complexes. We cannot exclude the possibility that CDK2 is involved in this process. In fact, Rb is exclusively mono-phosphorylated in early G1 phase by CDK4/CyclinD complexes and subsequently is hyper-phosphorylated by CDK2/CyclinE complexes in late G1 restriction point (Narasimha et al, 2014; Sanidas et al, 2019). Therefore, both CDK2 and CDK4 might be activated following p21 knockdown. However, double knockdown p21 and CDK4 reverses the p21 knockdown effects, while CDK2 remains active, probably because Rb is not mono-phosphorylated

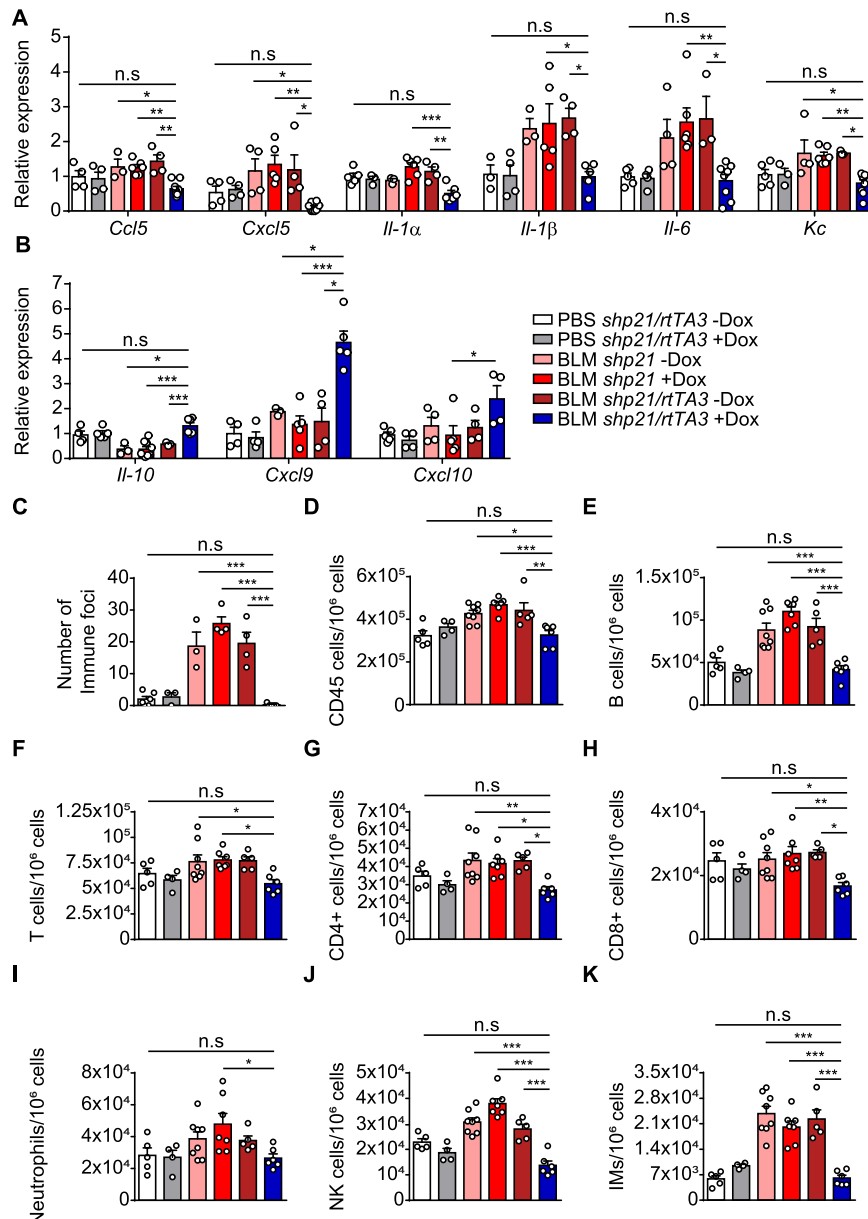

**Figure 6. p21 knockdown reduces inflammatory responses caused by exposure to bleomycin.**

*shp21* and *shp21/rtTA3* mice were administered once with bleomycin (BLM) or PBS vehicle and treated with Doxycycline (Dox) in the drinking water starting 10 days thereafter. Lungs were analyzed 21 days post BLM administration. (A, B) mRNA expression levels of the indicated SASP factors in *shp21/rtTA3* and *shp21* mice (BLM *shp21/rtTA3* –Dox vs. BLM *shp21/rtTA3* +Dox, *Ccl5* [$P = 0.0016$], *Cxcl5* [$P = 0.021$], *Il-1a* [$P = 0.003$], *Il-1β* [$P = 0.02$], *Il-6* [$P = 0.019$], *Kc* [$P = 0.024$], *Il-10* [$P = 0.012$], *Cxcl9* [$P < 0.0001$], *Cxcl10* [$P = 0.046$]). (C) Number of Immune foci in lung sections were quantified following H&E staining (BLM *shp21/rtTA3* –Dox vs. BLM *shp21/rtTA3* +Dox, $P < 0.0001$). (D–K) The flow-cytometry analysis of *shp21* and *shp21/rtTA3* mice lungs for number of: (D) immune cells (CD45 +), (E) B cells (CD45 + /B220 +), (F) T cells (CD45 + /CD3 +), (G) CD4 + T cells (CD45 + /CD3 + /CD4 +), (H) CD8 + T cells (CD45 + /CD3 + /CD8 +), (I) neutrophils (CD45 + /Ly6G + /CD11b +), (J) NK cells (CD45 + /Ly6G-/Nkp46 +) and (K) interstitial macrophages (IM's) (CD45 + /CD11c + /Siglec-F-/CD11b + /CD24 +) (BLM *shp21/rtTA3* –Dox vs. BLM *shp21/rtTA3* +Dox, CD45+ [$P = 0.008$], B cells, [$P = 0.0004$], T cells [$P = 0.025$], CD4 + T cells [$P = 0.016$], CD8 + T cells [$P = 0.011$], neutrophils [$P = 0.032$], NK cells [$P < 0.0001$], IM's [$P < 0.0001$]). Data information: Data were analyzed using one-way ANOVA. *$P < 0.05$. **$P < 0.005$. ***$P < 0.0005$. Data are presented as mean ± SEM (A, B, $n = 3–9$; C, $n = 3–6$; D–K, $n = 4–8$ independent repeats). Source data are available online for this figure.

by CDK4 and thus cannot be further hyper-phosphorylated and inactivated by CDK2.

We show that the p21-mediated regulation of the expression of ECM components in both senescent and non-senescent cells occurs in an Rb-dependent manner. Knockdown of p21 in senescent cells leads to the induction of apoptosis. Consequently, the observed

decrease in extracellular matrix (ECM) component production may be attributable to the loss of senescent cells. However, p21 also regulates ECM production in proliferating cells, where apoptosis does not occur. Therefore, it can be inferred that the reduction in ECM components is not due to cell death. Interestingly, the TGF-β pathway is downregulated following p21 knockdown in senescent

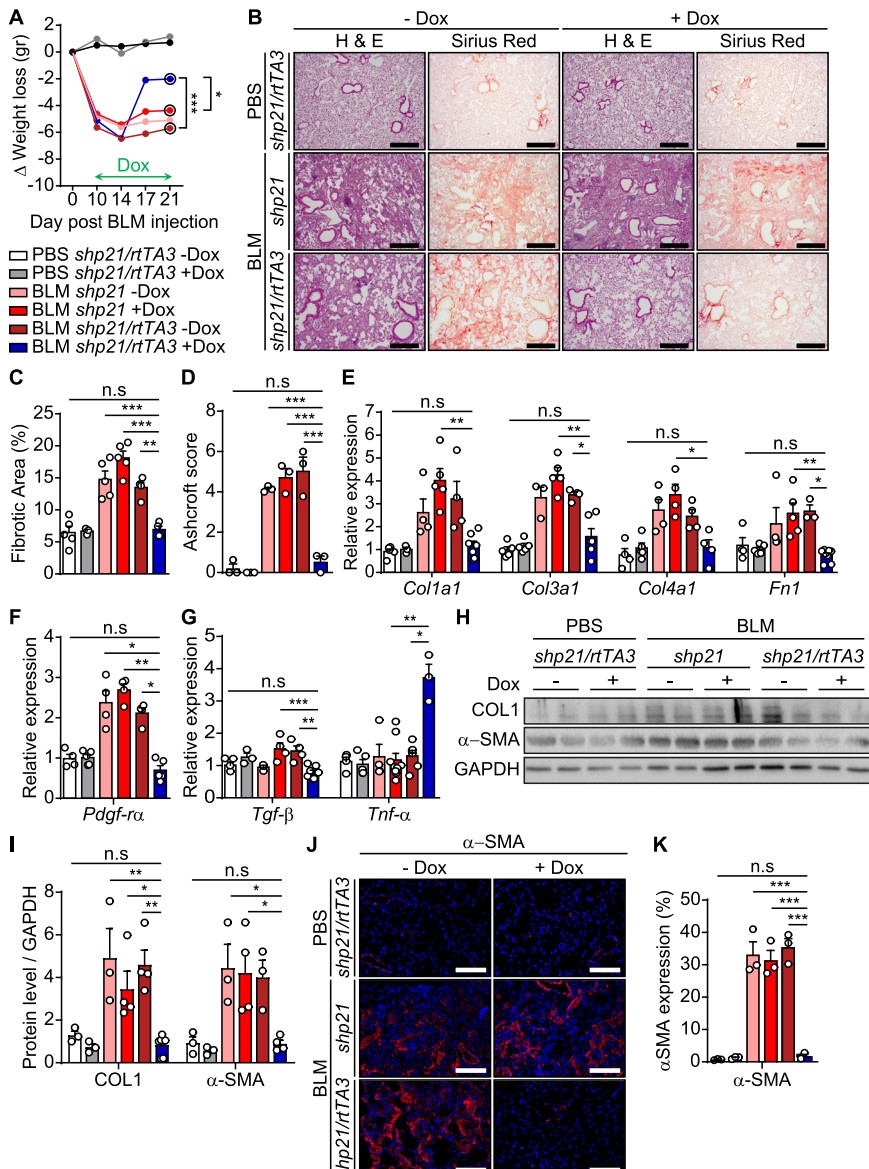

**Figure 7.  p21 knockdown alleviates bleomycin-induced lung fibrosis.**

*shp21* and *shp21/rtTA3* mice were administered once with bleomycin (BLM) or PBS vehicle and treated with Doxycycline (Dox) in the drinking water starting 10 days thereafter. Lungs were analyzed 21 days post BLM injection. (A) Body weight was monitored every 4 days and is presented as delta (Δ) from the starting weight (BLM *shp21/rtTA3* –Dox vs. BLM *shp21/rtTA3* +Dox, day 21 [P < 0.0001]). (B) H&E and Sirius Red staining of lung sections from *shp21/rtTA3* and *shp21* mice treated with BLM and Dox as described in (A). Scale bar: 200 μm. (C) Quantification of Sirius Red staining presented in (B) (BLM *shp21/rtTA3* –Dox vs. BLM *shp21/rtTA3* +Dox, P = 0.0028). (D) Mean Ashcroft scores of the mice lungs (BLM *shp21/rtTA3* –Dox vs. BLM *shp21/rtTA3* +Dox, P < 0.0001). (E) mRNA expression levels of collagen-1 (*Col1a1*), collagen-3 (*Col3a1*), collagen-4 (*Col4a1*) and Fibronectin-1 (*Fn1*) relative to PBS controls in the mice lungs (BLM *shp21* +Dox vs. BLM *shp21/rtTA3* +Dox, *Col1a1* [P = 0.0042], *Col3a1* [P = 0.0039], *Col4a1* [P = 0.036], *Fn1* [P = 0.043]). (F, G) mRNA expression levels of *Pdgf-ra* (F) and of *Tgf-β* and *Tnf-α* (G) relative to PBS controls in the mice lungs (BLM *shp21/rtTA3* +Dox, *Pdgf-ra* [P = 0.02], *Tgf-β* [P = 0.002], *Tnf-α* [P = 0.025]). (H) Western blot analysis of collagen-1 (COL1) and α-SMA in the mice lungs. (I) Quantification of protein levels of COL1 and α-SMA relative to GAPDH control (BLM *shp21/rtTA3* –Dox vs. BLM *shp21/rtTA3* +Dox, COL1 [P = 0.002], α-SMA [P = 0.023]). (J) IF staining of lung sections for α-SMA. Scale bar: 200 μm. (K) Quantification of α-SMA expression as presented in (J) (BLM *shp21/rtTA3* –Dox vs. BLM *shp21/rtTA3* +Dox, P < 0.0001). Data were analyzed using one-way ANOVA. *P < 0.05. **P < 0.005. ***P < 0.0005. Data are presented as mean ± SEM (A, n = 4–11; B, C, n = 3–5; D, n = 3; E, n = 4–5; F, n = 3–8; G, n = 3–9; H, I, n = 3–7; J, K, n = 3 independent repeats). Source data are available online for this figure.

cells (Yosef et al, 2017), and following p21 knockdown Smad-3 phosphorylation at the C-terminus (Serine 423/425) is decreased. Indeed, hypo-phosphorylated Rb can bind to Smad-3 and upregulate its transcriptional activity (Sturmlechner et al, 2021). Therefore, p21 knockdown leads to Rb phosphorylation, which

causes its detachment from Smad-3. Subsequently, Smad-3 is inhibited, leading to a decrease in ECM components expression. Alternatively, the activity of Smad-3 could be negatively regulated by the inhibitory Smad-7, which reduces TGF-β/Smad-3 signaling (Massague et al, 2005). Indeed, Smad-7 is upregulated in senescent

cells following p21 knockdown (Yosef et al, 2017). Therefore, the regulation of the ECM components expression following p21 knockdown could be mediated by several components of the TGF-β pathway.

The increase in p21 levels maintains the presence of senescent cells in pathological conditions in vivo. Although induction of p21 was shown to have anti-fibrotic effects in lung fibrosis pathology (Inoshima et al, 2004), other studies have shown that p21-positive senescent cells contribute to the progression of lung fibrosis and chronic obstructive pulmonary disease (COPD) (Liu and Liu, 2020; Schafer et al, 2017). Importantly, the presence of senescent cells following BLM administration in the lungs of $p21^{-/-}$ mice and *shp21/rtTA3* mice treated with Dox (group p21D) is diminished. This reduction might result from their decreased formation or induction of apoptosis in p21 deficient cells following DDR activation. Reduced fibrosis in $p21^{-/-}$ or p21 KD mice might result from cell-autonomous and cell non-autonomous effects. In a cell-autonomous manner, the inhibitory effect of p21 silencing on TGF-β signaling and ECM production directly leads to reduced collagen deposition and fibrosis. Alternatively, p21 silencing increases TNF-α production, which can antagonize the pro-fibrotic effect of TGF-β in a cell non-autonomous manner. Therefore, p21 inhibition not only leads to a reduction in the number of senescent cells, but also limits fibrotic processes.

Presently, in vivo shRNA technology allows to identify the effects of gene knockdown without the outcomes of the continuous absence of the gene of interest observed in knockout mice (Dow et al, 2012; Premsrirut et al, 2011). Here, we utilized this technology for the first time to study the effect of p21 knockdown in vivo in a lung fibrosis model. Our results showed that Dox-treated *shp21/rtTA3* mice (group p21D) exhibited lower fibrosis levels than other treatment groups. To exclude a possible bias of Dox treatment, we included a control group of single-transgenic *shp21* mice treated with Dox, which did not show any improvement in the lung fibrosis pathology following BLM administration. Furthermore, shRNA expression could present off-target effects, both sequence-dependent and sequence-independent. Nonetheless, to validate the role of p21 in vivo in promoting lung fibrosis, we also examined and compared the progression of the disease in $p21^{-/-}$ mice and observed similar results. Therefore, the phenotypes observed in our experiments are mediated by p21 itself, suggesting the central role of p21 in the regulation of the microenvironment of senescent cells and in fibrotic pathologies.

p21 limits tissue regeneration and promotes aging phenotypes (Baker et al, 2013; Choudhury et al, 2007). p21 knockout prolongs the lifespan of telomerase-deficient mice by rescuing the progenitor cells fitness (Choudhury et al, 2007). It also supports tissue regeneration in the liver (Marhenke et al, 2014), articular cartilage (Jablonski et al, 2021), bone (Premnath et al, 2017), and skin (Jiang et al, 2020). In addition, p21 is essential in modulating the secretome to affect the immunosurveillance of stressed cells (Sturmlechner et al, 2021). Therefore, we suggest that p21 silencing may not only limit the damage induced by the presence of senescent cells due to their elimination (Yosef et al, 2017), but also promote the recovery from tissue damage via modulation of the cellular microenvironment and reduction in the ECM production and inflammation in both proliferating and senescent cells. Overall, p21 promotes fibrosis and controls central molecular mechanisms regulating ECM expression and the viability of senescent cells. Therefore, inhibition of p21 is a plausible strategy for the efficient treatment of fibrotic and non-fibrotic age-related pathologies.

# Methods

**Reagents and tools table**

| Reagent/resource | Reference or source | Identifier or catalog number |
|---|---|---|
| **Experimental models: cell lines** | | |
| Human: BJ cells | ATCC | CRL-4001 |
| Human: IMR-90 cells | ATCC | CCL-186 |
| Mouse: WT (C57Bl/6) mouse lung fibroblast cells | This paper | N/A |
| Mouse: p21-/- (Cdkn1a<tm1Led >/J) mouse lung fibroblast cells | This paper | N/A |
| **Experimental models: organisms/strains** | | |
| Mouse: B6.129S6(Cg)-Cdkn1a<tm1Led >/J : p21-/- | Jackson Laboratory | Cat# 016565; RRID: IMSR_JAX:016565 |
| Mouse: B6N.FVB(Cg)-Tg(CAG-rtTA3)4288Slowe/J : CAGs-rtTA3 | Jackson Laboratory (Premsrirut et al, 2011) | Cat# 016532; RRID: IMSR_JAX:016532 |
| Mouse: TRE-tGFP-shCDKN1A : shp21 | Mirimus Inc., (Premsrirut et al, 2011) | Cat# C72-0530 |
| Mouse: TRE-tGFP-shCDKN1A x CAGs-rtAT3 : shp21 x rtAT3 | This paper | N/A |
| Mouse: C57BL/6 J : WT | Harlan | N/A |
| **Antibodies** | | |
| Rabbit polyclonal anti-p15 | Abcam | Cat# ab53034; RRID: AB_2078578 |
| Rabbit monoclonal anti-p16 | Abcam | Cat# ab108349; RRID: AB_10858268 |
| Rabbit polyclonal anti-p19 | Abcam | Cat# ab80; RRID: AB_306197 |
| Rabbit monoclonal anti-α-Smooth Muscle Actin (αSMA) | Cell Signaling Technology | Cat# 19245S; RRID: AB_2734735 |
| Rabbit polyclonal anti-phospho-Histone H2A.X (Ser139) (γH2A.X) | Cell Signaling Technology | Cat# 2577S; RRID: AB_2118010 |

| Reagent/resource | Reference or source | Identifier or catalog number |
|---|---|---|
| Rabbit monoclonal anti-CDK4 | Cell Signaling Technology | Cat# 12790S; RRID: AB_2631166 |
| Mouse monoclonal Phospho-ATM (Ser1981) (p-ATM) | Cell Signaling Technology | Cat # 4526 T; RRID: AB_2062663 |
| Rabbit polyclonal Phospho-Rb (Ser780) (p-Rb) | Cell Signaling Technology | Cat # 9307S; RRID: AB_330015 |
| Rabbit monoclonal Phospho-Smad-3 (Ser423/425) | Cell Signaling Technology | Cat # 9520S; RRID: AB_2193207 |
| Rabbit monoclonal RB1 | Cell Signaling Technology | Cat # 9313S; RRID: AB_1904119 |
| Rabbit monoclonal Smad-3 | Cell Signaling Technology | Cat # 9523S; RRID: AB_2193182 |
| Mouse monoclonal Glyceraldehyde-3-Phosphate Dehydrogenase (GAPDH) | Merck Millipore | Cat# MAB374; RRID: AB_2107445 |
| Rabbit polyclonal Collagen Type I | Rockland Immunochemicals | Cat# 600-401-103-0.5; RRID: AB_217595 |
| Mouse monoclonal p21 | Santa Cruz Biotechnology | Cat# sc-6246; RRID: AB_628073 |
| Mouse monoclonal β-Actin | Sigma-Aldrich | Cat# A5441; RRID: AB_476744 |
| Mouse polyclonal p53 hybridoma | M. Oren, Weizmann Institute of Science | N/A |
| Rat monoclonal PerCP/Cy5.5 anti-mouse/human CD45R/B220 (clone RA3-6B2) | BioLegend | Cat# 103236; RRID: AB_893354 |
| Rat monoclonal APC anti-mouse CD4 (clone GK1.5) | BioLegend | Cat# 100412; RRID: AB_312697 |
| Rat monoclonal FITC anti-mouse CD8a (clone 53-6.7) | BioLegend | Cat# 100706; RRID: AB_312745 |
| Rat monoclonal PerCP/Cy5.5 anti-mouse/human CD11b (clone M1/70) | BioLegend | Cat# 101228; RRID: AB_893232 |
| Hamster monoclonal APC anti-mouse CD11c (clone N418) | BioLegend | Cat# 117310; RRID: AB_313779 |
| Rat monoclonal PE/Cy7 anti-mouse CD24 (clone M1/69) | BioLegend | Cat# 101822; RRID: AB_756048 |
| Rat monoclonal Pacific Blue anti-mouse CD45 (clone 30-F11) | BioLegend | Cat# 103126; RRID: AB_493535 |
| Rat monoclonal FITC anti-mouse CD45 (clone 30-F11) | BioLegend | Cat# 103108; RRID: AB_312973 |
| Hamster monoclonal PE anti-mouse CD103 (clone 2E7) | BioLegend | Cat# 121406; RRID: AB_1133989 |
| Rat monoclonal APC anti-mouse Ly6G (clone 1A8) | BioLegend | Cat# 127614; RRID: AB_2227348 |
| Rat monoclonal APC/Cyanine7 anti-mouse MHC-II (clone M5/114.15.2) | BioLegend | Cat# 107627; RRID: AB_1659252 |
| Rat monoclonal PE anti-mouse CD335 (NKp46) (clone 29A1.4) | BioLegend | Cat# 137604; RRID: AB_2235755 |
| Hamster monoclonal PE anti-mouse TCRβ chain (clone H57-597) | BioLegend | Cat# 109208; RRID: AB_313431 |
| Rat monoclonal BV421 Rat Anti-Mouse Siglec-F (clone E50-2440) | BD Biosciences | Cat# 562681; RRID:AB_2722581 |
| Rabbit monoclonal anti-p16 | Abcam | Cat# ab211542; RRID: AB_2891084 |
| Rat monoclonal anti-p21 | Abcam | Cat# ab107099; AB_10891759 |
| Cy3 AffiniPure Goat Anti-Rabbit IgG (H + L) antibody | Jackson ImmunoResearch | Cat# 111-165-144; RRID: AB_2338006 |
| Cy3 AffiniPure Donkey Anti-Rat IgG (H + L) antibody | Jackson ImmunoResearch | Cat# 712-165-153; RRID:AB_2340667 |
| Cy5 AffiniPure Goat Anti-Rabbit IgG (H + L) antibody | Jackson ImmunoResearch | Cat# 111-175-144; RRID: AB_2338013 |
| Oligonucleotides and other sequence-based reagents | | |
| siRNA targeting sequence: human CDKN1A (p21) | Dharmacon | Cat# L-003471-00-0020; Gene ID:1026 |
| siRNA targeting sequence: human CDK4 | Dharmacon | Cat# L-003238-00-0020; Gene ID:1019 |
| siRNA targeting sequence: human CCND1 (CyclinD1) | Dharmacon | Cat# L-003210-00-0020; Gene ID:595 |
| siRNA targeting sequence: human Rb1 | Dharmacon | Cat# L-003296-02-0020; Gene ID: 5925 |
| siRNA targeting sequence: non-targeting siRNA pool | Dharmacon | Cat# D-001810-10-20 |
| qPCR primers for human, see below | This paper | N/A |
| qPCR primers for mouse, see below | This paper | N/A |
| Chemicals, enzymes, and other reagents | | |
| Liberase TM Research Grade | Roche | Cat# 5401129001; CAS: 10035-04-8 |
| 1X antibiotic/antimycotic | Thermo Fisher Scientific | Cat# 15420-096 |
| Etoposide | Sigma-Aldrich | Cat# E1383; CAS: 33419-42-0 |

| Reagent/resource | Reference or source | Identifier or catalog number |
|---|---|---|
| Abemaciclib | Pubchem | Cat# LY2835219; CAS: 1231929-97-7 |
| NP40 | Sigma-Aldrich | Cat# 1302; CAS: 9002-93-1 |
| Doxycycline hyclate (Dox) | Sigma-Aldrich | Cat# D9891; CAS: 24390-14-5 |
| Bleomycin sulfate (BLM) | Sigma-Aldrich | Cat# B2434; CAS: 9041-93-4 |
| Collagenase type 4 | Worthington Biochemical Corporation | Cat# LS004189 |
| DNaseI | Roche | Cat# 04536282001 |
| Glutaraldehyde 25% | Electron Microscopy Sciences | Cat# 50-261-96 |
| X-Gal 40x solution | Inalco Pharmaceuticals | Cat# 1758-0300; CAS: 7240-90-6 |
| Potassium ferrocyanide | Merck Millipore | Cat# 104973; CAS: 13746-66-2 |
| Potassium ferricyanide | Sigma-Aldrich | Cat# P3289; CAS: 14459-95-1 |
| Nuclear Fast Red | Sigma-Aldrich | Cat# N3020; CAS: 6409-77-4 |
| DMEM medium | Thermo Fisher Scientific | Cat# 11965-092 |
| Penicillin/Streptomycin | Biological Industries | Cat# 03-031-1B |
| Fetal bovine serum (FBS) | Thermo Fisher Scientific | Cat# 10270, Lot# 2289265 |
| DMEM/F12 medium | Thermo Fisher Scientific | Cat# 11330-032 |
| Fetal bovine serum (FBS) | Thermo Fisher Scientific | Cat# 26140-079 |
| EMEM medium | ATCC | Cat# 30-2003 |
| Dynabeads™ Protein G | Thermo Fisher Scientific | Cat# 10003D |
| DharmaFECT formulation 1 reagent | Dharmacon | Cat# T-2001-03 |
| RNeasy mini kit | Qiagen | Cat# 74104 |
| NucleoSpin RNA Mini kit | Macherey-Nagel | Cat# 740955.50 |
| Random hexamers | Thermo Fisher Scientific) | Cat# N8080127 |
| Platinum SYBR Green qPCR SuperMix | Life Technologies | Cat# 11744-500 |
| ImmobilonP membranes | Millipore | Cat# IPVH00010 |
| 5% bovine serum albumin (BSA) | Sigma-Aldrich | Cat# A7906 |
| SuperSignal West Pico PLUS chemiluminescent substrate | Thermo Fisher Scientific | Cat# 34579 |
| SuperSignal West Femto maximum sensitivity substrate | Thermo Fisher Scientific | Cat# 34095 |
| RMPI medium | Thermo Fisher Scientific | Cat# 11875-093 |
| Red blood cell lysis buffer | Sigma-Aldrich | Cat# R7757 |
| Antigen unmasking solution, tris-based (pH=9) | Vector Laboratories | Cat# H-3301-250 |
| CAS-Block™ histochemical reagent | Thermo Fisher Scientific | Cat# 008120 |
| DAPI | Sigma-Aldrich | Cat# D9542; CAS: 28718-90-3 |
| Fluormount-G | Southern Biotech | Cat# 0100-01 |
| EUKITT mounting solution | Sigma-Aldrich | Cat# 03989; CAS: 25608-33-7 |
| **Software and algorithms** | | |
| StepOnePlus Real-Time PCR System | Applied Biosystems | https://www.thermofisher.com/il/en/home/technical-resources/software-downloads/StepOne-and-StepOnePlus-Real-Time-PCR-System.html |
| Image Lab software | Bio-Rad Laboratories | https://www.bio-rad.com/en-il/product/image-lab-software?ID=KRE6P5E8Z |
| FlowJo v10 software | BD Biosciences | https://www.flowjo.com/solutions/flowjo/downloads |
| ImageJ software | NIH | https://imagej.net/ |
| CellP software | Diagnostic Instruments | N/A |
| Prism | GraphPad | https://www.graphpad.com/features |
| R software (version 4.3.1) | R project | https://cran.r-project.org/bin/windows/base/old/ |

| Reagent/resource | Reference or source | Identifier or catalog number |
|---|---|---|
| **Other** | | |
| **Deposited data** | | |
| p21-bound proteins detected by mass spectrometry | This paper | N/A |
| **qPCR primers for human** | | |

| Primer | Forward sequence | Reverse sequence |
|---|---|---|
| *GAPDH* | 5′-GACAGTCAGCCGCATCTTC-3′ | 5′-CGTTGACTCCGACCTTCAC-3′ |
| *HPRT* | 5′-TGACACTGGCAAAACAATGCA-3′ | 5′-GGTCCTTTTCACCAGCAAGCT-3′ |
| *CCND1* | 5′-ATGTTCGTGGCCTCTAAGATGA-3′ | 5′-CAGGTTCCACTTGAGCTTGTTC-3′ |
| *CDK4* | 5′-TCGAAAGCCTCTCTTCTGTG-3′ | 5′-TACATCTCGAGGCCAGTCAT-3′ |
| *COL1A1* | 5′-CATGTCTGGTTCGGCGAGAG-3′ | 5′-GCAGGAAGGTCAGCTGGATG-3′ |
| *COL3A1* | 5′-AGGAGCTAACGGTCTCAGTG-3′ | 5′-ACCATCTGATCCAGGGTTTC-3′ |
| *COL4A1* | 5′-GAAGTGCGCCATTCATCGAG-3′ | 5′-CTTCAAGGTGGACGGCGTAG-3′ |
| *FN1* | 5′-TGGACATGCATTGCCTACTC-3′ | 5′-CATGACGCTTGTGGAATGTG-3′ |
| *p21* | 5′-GCTGCGTTCACAGGTGTTTC-3′ | 5′-CATGGGTTCTGACGGACATC-3′ |
| *Rb* | 5′-TCCAGACCCAGAAGCCATTG-3′ | 5′-CTGGGTGCTCAGACAGAAGG-3′ |
| **qPCR primers for mouse** | | |

| Primer | Forward sequence | Reverse sequence |
|---|---|---|
| *Gapdh* | 5′-TGCACCACCAACTGCTTAGC-3′ | 5′-GGCATGGACTGTGGTCATGAG-3′ |
| *Hprt* | 5′-GATTAGCGATGATGAACCAGGTT-3′ | 5′-CCTCCCATCTCCTTCATGACA-3′ |
| *Ccl5* | 5′-GCCCTCACCATCATCCTCAC-3′ | 5′-ATCCCATTTTCCCAGGACC-3′ |
| *Col1a1* | 5′-AGGATCTCCTGGTGCTGATG-3′ | 5′-GGAAGCCTCTTTCTCCTCTC-3′ |
| *Col3a1* | 5′-AACTGGAGCACGAGGTCTTG-3′ | 5′-ATTATGGCCACTGGCTCCTG-3′ |
| *Col4a1* | 5′-TATCTCTGGGGACAACATCCG-3′ | 5′-CATCTCGCTTCTCTCTATGGTG-3′ |
| *Cxcl5* | 5′-GTTCCATCTCGCCATTCATGC-3′ | 5′-GCGGCTATGACTGAGGAAGG-3′ |
| *Cxcl9* | 5′-TCTTCCTGGAGCAGTGTGG-3′ | 5′-TCCGGATCTAGGCAGGTTT-3′ |
| *Cxcl10* | 5′-CCATCAGCACCATGAACC-3′ | 5′-TCCGGATTCAGACATCTC-3′ |
| *Fn1* | 5′-CGAGGTGACAGAGACCACAA-3′ | 5′-CTGGAGTCAAGCCAGACACA-3′ |
| *Ifn-γ* | 5′-CATGGCTGTTTCTGGCTGTTACTG-3′ | 5′-GTTGCTGATGGCCTGATTGTCTTT-3′ |
| *Il-1-α* | 5′-TCAACCAAACTATATATATCAGGATGTGG-3′ | 5′-CGAGTAGGCATACATGTCAAATTTTAC-3′ |
| *Il-1-β* | 5′-GGAGAACCAAGCAACGACAAAATA-3′ | 5′-TGGGGAACTCTGCAGACTCAAAC-3′ |
| *Il-6* | 5′-AGACAAAGCCAGAGTCCTTC-3′ | 5′-TGCCGAGTAGATCTCAAAGT-3′ |
| *Il-10* | 5′-CTGGGTGAGAAGCTGAAGAC-3′ | 5′-ACTCTTCACCTGCTCCACTG-3′ |
| *Kc* | 5′-AAGAATGGTCGCGAGGCTTG-3′ | 5′-TGCCATCAGAGCAGTCTGTC-3′ |
| *p15* | 5′-CCACCCTTACCAGACCTGTG-3′ | 5′-AGGCGTCACACACATCCAG-3′ |
| *p16* | 5′-TTGGGCGGGCACTGAATCTC-3′ | 5′-AGTCTGTCTGCAGCGGACTC-3′ |
| *p21* | 5′-GACAAGAGGCCCAGTACTTC-3′ | 5′-GCTTGGAGTGATAGAAATCTGTC-3′ |
| *Pdgf-Rα* | 5′-TCTCCAGCGACAAGGAACAG-3′ | 5′-CTGTGGATGCTCCCATTACC-3′ |
| *shp21* | 5′-AAGCCACAGATGTATTAAGACAC-3′ | 5′-CACCCTGAAAACTTTGCCCC-3′ |
| *Tgf-β* | 5′-CACCGGAGAGCCCTGGATA-3′ | 5′-TGTACAGCTGCCGCACACA-3′ |
| *Tnf-α* | 5′-CCACGCTCTTCTGTCTACTG-3′ | 5′-GATGAGAGGGAGGCCATTTG-3′ |

## Experimental model and subject details

### Cell culture

Human BJ and IMR-90 fibroblasts were obtained from the American Type Culture Collection (ATCC) and placed in a low oxygen (5% $CO_2$, 3% $O_2$) incubator at 37 °C. The cells were maintained in DMEM (11965-092, Thermo Fisher Scientific) supplemented with 2 mM L-glutamine, 100 units/ml of penicillin, 100 mg/ml of streptomycin (03-031-1B, Biological Industries) and 10% fetal bovine serum (FBS) (10270, Thermo Fisher Scientific). Mouse lung fibroblasts were isolated according to standard procedures (Seluanov et al, 2010). In short, lungs of WT or $p21^{-/-}$ mice were extracted to cold PBS, cut to 1 mm pieces and incubated at 37 °C for 30 min, with 10 ml of DMEM/F12 media (11330-032, Invitrogen) with 0.14 Wunsch units/mL Liberase TM Research Grade (05401127001, Roche), and 1× antibiotic/antimycotic (15420-096, Invitrogen). Then, the lung pieces were supplemented with 30 ml DMEM/F12 media with 15% FBS (26140-079, Thermo Fisher Scientific), 1× antibiotic/antimycotic, centrifuged twice, and placed in a 10-cm plate in a low oxygen (5% $CO_2$, 3% $O_2$) incubator at 37 °C. The extracted fibroblasts were then maintained in EMEM (30-2003, ATCC) supplemented with 2 mM L-glutamine, 100 units/ml of penicillin, 100 mg/ml of streptomycin, and non-heat inactivated 15% FBS (26140-079, Thermo Fisher Scientific).

### Mice

$p21^{-/-}$ mice [B6.129S6(Cg)-Cdkn1a<tm1Led > /J] and *CAGs-rtTA3* [B6N.FVB(Cg)-Tg(CAG-rtTA3)4288Slowe/J]] mice were obtained from the Jackson Laboratory (#016565 and #016532, respectively) and maintained on C57Bl/6 background. *shp21* mice (*TRE-tGFP-shCDKN1A*) were obtained from Mirimus Inc. (C72-0530) and maintained on C57Bl/6 background. The transgene includes a targeting construct for shRNA targeting p21, linked to a GFP fluorescent reporter, all under a TRE promoter. *CAG-rtTA3* and *shp21* mice were crossed to produce *shp21/rtAT3* (*CAG-rtTA3/tet-shCDKN1A*) transgenic mice.

## Method details

### Cellular senescence induction

DNA damage-induced senescence (DIS) was induced by treatment with etoposide (E1383, Sigma-Aldrich) at a concentration of 20 μM (for mouse lung fibroblasts) or 50 μM (for BJ and IMR-90) for 48 h. By 7 days post-treatment, the cells had acquired the senescence phenotype (Yosef et al, 2017).

### CDK4 inhibition

Target mouse lung fibroblast cells were plated in 6-well plates. When indicated, CDK4 inhibitor Abemaciclib (LY2835219, Pubchem) was added to the cells at a concentration of 1 μM for 72 h.

### Immunoprecipitation

Immunoprecipitation of p21 was carried out by using either proliferating or DIS BJ cells. Cell pellets were suspended in lysis buffer (150 mM HEPES, 150 mM NaCl, 0.1% NP40 [13021, Sigma-Aldrich]) for 40 min on 4 C and then sonicated using a homogenizer. After centrifugation, the supernatant was mixed with Dynabeads™ Protein G (10003D, Thermo Fisher Scientific) and a p21 antibody, and the mixture was left at slow shaking for 16 h at 4 °C. The beads were washed five times (2× with TBS/T

[with Tween 0.05%], 3× with wash buffer [TBS/T with 150 mM Hepes, 150 mM NaCl]). The suspension was boiled for 5 min and loaded to a 12.5% SDS–PAGE. In addition, samples were also analyzed by mass spectrometry proteomics in order to identify proteins that are bound to p21.

### siRNA

Target human BJ or IMR-90 fibroblasts were plated in 12-well plates. Cells were transfected with Dharmacon ON-TARGETplus SMARTpool small interfering RNA (siRNA) pools targeting human CDKN1A (p21) (L-003471-00-0020), human CDK4 (L-003238-00-0020), human CCND1 (CyclinD1) (L-003210-00-0020), human Rb1 (L-003296-02-0020) or, as a control, with non-targeting siRNA pool (D-001810-10-20), using DharmaFECT formulation 1 reagent (T-2001-03, Dharmacon) according to the instructions of the manufacturer.

### Quantitative RT-PCR

Total RNA from BJ, IMR-90, and MLF cells was extracted using an RNeasy mini kit (74104, QIAGEN), followed by DNase-I treatment. Total RNA from mice lungs was extracted using NucleoSpin RNA Mini kit (740955.50, Macherey-Nagel). cDNA was produced using random hexamers (N8080127, Thermo Fisher Scientific). The cDNA samples were amplified using Platinum SYBR Green qPCR SuperMix (11744-500, Life Technologies) in a StepOnePlus Real-Time PCR System (Applied Biosystems). Relative expression was normalized using the expression levels of GAPDH or HPRT. Primer sequences for human and mouse can be found in the Regents and Tools table.

### Immunoblotting

Cell lysates (15–30 mg of protein) were resolved by 12.5% SDS–PAGE and transferred onto ImmobilonP membranes (IPVH00010, Millipore). After blocking of the membranes with 5% bovine serum albumin (BSA) (A7906, Sigma-Aldrich) in TBST (Tris-buffered saline with 0.01% Tween-20) for 1 h, they were probed with antibodies against p15 (ab53035), p16 (ab108349), p19 (ab80) (all from Abcam), α-SMA (#19245), γH.2AX (Ser139, #2577), CDK4 (#12790), phospho-ATM (Ser1981, #4526), phospho-Rb (Ser780, #9307), phospho-Smad-3 (Ser423/425, #9520), Rb (#9313), Smad-3 (#9523) (all from Cell Signaling Technology), GAPDH (MAB374, Millipore), collagen-1 (600-401-103-0.5, Rockland Immunochemicals), p21 (sc-6246, Santa Cruz), β-actin (A5441, Sigma-Aldrich) and p53 (mix of DO-1 and PAb1801, kindly provided by M. Oren, Weizmann Institute of Science). The blots were developed using either SuperSignal West Pico PLUS chemiluminescent substrate (#34579) or SuperSignal West Femto maximum sensitivity substrate (#34095) (both from Thermo Fisher Scientific). The blots were analyzed using the Image Lab software (Bio-Rad Laboratories).

### Mice procedures

**shCDKN1A expression induction by Dox administration**: Female and male *shp21/rtAT3* mice received Dox (D9891, Sigma-Aldrich) at a concentration of 2 mg/ml in the drinking water at 8 weeks of age for induction of the *shp21* transgene expression. *shp21* single-transgene sibling control mice received Dox for the same period.
**p21 induction by irradiation**: Female *shp21/rtAT3* mice received Dox at a concentration of 2 mg/ml in the drinking water at 8 weeks

of age for activation of the *shp21* transgene. Three days following transgene activation, mice were irradiated with 8 Gy for induction of DNA damage and senescence. Mice were sacrificed 16 h following irradiation, and lungs were frozen in OCT solution for cryosectioning, or homogenized for RNA and protein extraction.

**Lung fibrosis induction**: *shp21/rtAT3* and *shp21* 8-week-old female mice were administered with bleomycin (BLM) (1.5 u/kg) (B2434, Sigma-Aldrich) or PBS vehicle as a control by intra-tracheal installation. The mice initial weighs ranged between 20 and 25 g, and were measured every 4 days. Ten days post BLM injection mice were treated with Dox for 11 days to activate *shp21* expression. Mice were sacrificed 21 days following BLM administration and their lungs were paraffin-embedded for immunohistology, frozen in OCT solution for cryosectioning and SA-β-gal stains, homogenized for single-cell FACS analysis or homogenized for RNA and protein extraction. Wild-type (WT) or p21 knockout (*p21⁻/⁻*) 8-week-old female mice were administered with BLM (1.5 u/kg) or PBS vehicle as a control by intra-tracheal installation and sacrificed 3, 10, or 21 days thereafter. The lungs of the mice were paraffin-embedded for immunohistology, frozen in OCT solution for cryosectioning and SA-β-gal stains, homogenized for single-cell FACS analysis or homogenized for RNA and protein extraction. The Weizmann Institute of Science Animal Care and Use Committee (IACUC) approved all procedures described in this work.

### Single-cell lung homogenate preparation and FACS analysis

The lungs of euthanized mice were removed, washed in RPMI medium (11875-093 Thermo Fisher Scientific), minced, and incubated at 37 °C for 45 min in RPMI medium containing 1 mg/ml collagenase type 4 (LS004189, Worthington), and 0.02 mg/ml DNaseI (04536282001, Roche). Lung cell suspensions were pushed through a 100-μm cell strainer and spun, and red blood cells were lysed with red blood cell lysis buffer (R7757, Sigma-Aldrich). Cells from whole lungs were collected, washed twice with FACS buffer, immunolabeled with antibodies against B220 (PerCP/Cy5.5, #103236), CD4 (APC, #100412), CD8a (FITC, #100706), CD11b (PerCP/Cy5.5, #101228), CD11c (APC, #117310), CD24 (PE-CY7, #101822), CD45 (PB, #103126 or FITC, #103108), CD103 (PE, #121406), Ly6G (APC, #127614), MHC-II (APC-CY7, #107627), NKp46 (PE, #137604), TCRb (PE, #109208) (all from BioLegend) and Siglec-F (PB, #562681, BD Biosciences). The cells were run in a LSR II Flow Cytometer (BD Biosciences) and analyzed using the FlowJo v10 software (BD Biosciences). The gating strategy for the immune subsets was performed as follows: neutrophils (CD45 + /Ly6G + /CD11b + ), NK cells (CD45 + /NKp46 + ), CD3 (CD45 + /TCRb + ), CD4 (CD45 + /TCRb + /CD4 + ), CD8 (CD45 + /TCRb + /CD8 + ), B cells (CD45 + /B220 + ) and interstitial macrophages (CD45 + / MHC-II + / CD11c + / Siglec-F- / CD11b + / CD24-).

### Histological analysis

Immunofluorescence (IF) was performed on 4-μm paraffin sections according to standard procedures. Sections were deparaffinized and rehydrated in an ethanol series. Antigen retrieval (H-3301-250, Vector Laboratories) was performed in a boiling water bath, and sections were blocked for non-specific binding with CAS-Block solution (008120, Thermo Fisher Scientific). Primary antibodies recognizing p15 (1:100; ab53034), p16 (1:100; ab211542), p21 (1:100; ab107099) (all from Abcam), α-SMA (1:500; #19245),

cleaved-Caspase 3 (1:400; #9661) and γH.2AX (1:100; Ser139, #2577) (all from Cell Signaling Technology) were applied overnight at 4 °C. p15, p16, α-SMA, cleaved-Caspase 3 and γH.2AX sections were incubated with Cy3 anti-rabbit antibody (111-165-144, Jackson ImmunoResearch), p16 sections were incubated with Cy5 anti-rabbit antibody (11-175-144, Jackson ImmunoResearch) and p21 sections were incubated with Cy3 anti-rat antibody (712-165-153, Jackson ImmunoResearch) for 90 min in a humidity chamber. Sections were counterstained by DAPI (D9542, Sigma-Aldrich) and then mounted using Fluormount (0100-01, Southern Biotech). Stained sections were examined and photographed with a fluorescence microscope (Olympus). All the images were quantified using NIH ImageJ software (http://rsb.info.nih.gov/ij/). For p15, p16, p21, cleaved-Caspase 3 and γH.2AX staining we counted positive cells in each field and for α-SMA staining we measured the percentage of positive staining in each field.

Paraffin-embedded tissue sections (4-μm) were stained with hematoxylin–eosin (H&E) for routine examination or with Sirius Red for visualization of fibrotic deposition. The sections were examined and photographed with a bright-field microscope (Olympus). These images were then quantified using NIH ImageJ software. We calculated the amount of fibrotic tissue relative to the basal amount of Sirius Red staining present in control mice lungs. Fibrosis severity was also evaluated in H&E-stained sections according to Ashcroft scale (Ashcroft et al, 1988).

For SA-β-gal staining, 14-μm cryosections of OCT embedded mouse lungs were fixed in 0.5% glutaraldehyde (50-261-96, Electron Microscopy Sciences) for 15 min, washed with PBS supplemented with 1 mM MgCl₂ in PBS at pH 5.5, and incubated for 6 h in X-Gal staining solution (1 mg/mL X-Gal (1758-0300, Inalco Pharmaceuticals), 5 mM potassium ferrocyanide (104973, Merck Millipore), 5 mM potassium ferricyanide (P3289, Sigma-Aldrich) and 1 mM MgCl₂ in PBS at pH 5.5). Sections were counterstained with Nuclear Fast Red (N3020, Sigma-Aldrich), dehydrated and mounted with Eukitt mounting solution (03989, Sigma-Aldrich). Sections were visualized using an Olympus microscope, and images were analyzed using CellP software (Diagnostic Instruments).

### Analysis of p21 expression from published scRNA-seq data from a bleomycin-induced lung injury study

For this analysis, data was handled mainly with Seurat package for single-cell RNA sequencing analyses, using the R software (version 4.3.1). Raw sequencing files of whole lung specimens, originally published by Strunz et al (Strunz et al, 2020), were downloaded from the Gene Expression Omnibus with the accession code GSE141259 (Strunz et al, 2020).

For the ECM components analysis, data was filtered to include only samples from day 10, 14, and 21 post BLM administration. Then, "Col1a1", "Col3a1", "Col4a1", "Fn1" genes were selected for further analysis, and were defined as "ECM components". A UMAP plot was created to present the clustering of cells based on cell type in the lung. Based on this clustering, 3 UMAP plots were created to present ECM components expression, p21 expression, and the combined expression of ECM components and p21. For the creation of the violin plots, "VlnPlot" function was called to display "ECM components" expression count values of the following clusters: AM's, monocytes (-AM's), T cells, AT2 cells, epithelial cells (-AT2), endothelial cells and stromal cells.

Analysis of alveolar macrophages and fibroblasts (stromal) cells was done separately. For the creation of the bar plots, "ggplot2" and "ggpubr" packages were used for displaying cell counts in each mouse and comparing means between BLM and PBS lung-infused mice using Wilcoxon non-parametric test. Violin plots displaying Collagen producing genes expression in each day were created for each cell type using "VlnPlot".

For p21 expression pattern, data was filtered to include only samples from day 3, 10, 14, and 21 post BLM administration. Then, counts were log normalized, scaled and processed for dimensionality determination. Clustering the cells was done by "FindNeighbors" with 15 dimensions, and "FindClusters" with resolution value of 0.8. Predictions of cell types was done using "SingleR" package against lung reference from the Tabula Muris database. For the creation of the violin plots, "VlnPlot" function was called to display "p21" count values of the 5 most abundant clusters, namely (1) Epithelial cells, (2) Endothelial cells, (3) Stromal cells, (4) Type II Pneumocytes and (5) White blood cells.

### Quantification and statistical analysis

Data are presented as means ± SEM. Statistical significance was determined using Student's *t* test or one-way ANOVA. A *P* value of <0.05 was considered significant. All statistical analysis was performed using the Prism software (GraphPad).

## Data availability

No large-scale data amenable to data repository deposition were generated in this study.

The source data of this paper are collected in the following database record: biostudies:S-SCDT-10_1038-S44318-024-00246-7.

## Peer review information

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

## Acknowledgements

The authors thank Y Levin for helping analyze the MS data; M Oren for anti-p53 antibodies; E Hagai and G Levi-Cohen from the Flow Cytometry unit for helping with cell analysis; and all the members of Krizhanovsky laboratory for stimulating discussions. This work was supported by grants to VK from the European Research Council under the European Union's H2020 (856487), from the Israel Science Foundation (2633/17; 1626/20), Israel Ministry of Health 3-15100, DFG - CRC 1506 "Aging at Interfaces", Weizmann—Belle S and Irving E Meller Center for the Biology and Aging, and Sagol Institute for Longevity Research (given to VK). VK is an incumbent of The Georg F Duckwitz Professorial Chair and Shimon and Golde Picker—Weizmann Award.

## Author contributions

**Nurit Papismadov**: Conceptualization; Data curation; Formal analysis; Investigation; Methodology; Writing—original draft; Writing—review and editing. **Naama Levi**: Data curation. **Lior Roitman**: Methodology. **Amit Agrawal**: Methodology. **Yossi Ovadya**: Methodology. **Ulysse Cherqui**: Methodology. **Reut Yosef**: Methodology. **Hagay Akiva**: Methodology. **Hilah Gal**: Writing—review and editing. **Valery Krizhanovsky**: Conceptualization; Writing—original draft; Project administration; Writing—review and editing.

Source data underlying figure panels in this paper may have individual authorship assigned. Where available, figure panel/source data authorship is listed in the following database record: biostudies:S-SCDT-10_1038-S44318-024-00246-7.

## Disclosure and competing interests statement

VK is an author of patents on senolytics and senolytic approaches. None of these influenced the work conducted or the analysis of data presented in this manuscript.

