## [Peer Review File · The EMBO Journal]

p21 regulates expression of ECM components and promotes pulmonary fibrosis via CDK4 and Rb

Nurit Papismadov, Naama Levi, Lior Roitman, Amit Agrawal, Yossi Ovadya, Ulysse Cherqui, Reut Yosef, Hagay Akiva, Hilah Gal, and Valery Krizhanovsky

Corresponding author: Valery Krizhanovsky (valery.krizhanovsky@weizmann.ac.il)

Review Timeline:

Submission Date:	21st May 24
Editorial Decision:	10th Jul 24
Editor's Correspondence:	11th Jul 24
Editor's Correspondence:	16th Jul 24
Revision Received:	5th Aug 24
Accepted:	5th Sep 24

Editor: Daniel Klimmeck

Transaction Report:

Please note that the manuscript was previously reviewed at another journal and the reports were taken into account in the decision making process at The EMBO Journal. Since the original reviews are not subject to EMBO Press' transparent review process policy, the reports and author response cannot be published.

Dear Dr Krizhanovsky,

Thank you again for the submission of your amended manuscript (EMBOJ-2024-117941) to The EMBO Journal. Please accept again my sincere apologies for the unusual protraction due to delayed expert input. We have carefully assessed your manuscript and the point-by-point response provided to the referee concerns that were raised during review at a different journal. In addition, and as mentioned before, we decided to involve an arbitrating expert to evaluate the revised version of your work, with respect to technical robustness, conceptual advance and overall suitability of your work for publication in The EMBO Journal.

As you will see from his/her comment enclosed below, the advisor is broadly in favour of the work stating the interest and value of your results and s/he is supportive of publication at The EMBO Journal, pending minor revision.

We are thus pleased to inform you that we can offer to swiftly move forward towards acceptance of this work at The EMBO Journal, pending minor revision of the following remaining issues, which need to be adjusted in a re-submitted version.

Based on the positive views of the advisor together with our own assessment, we decided to proceed with publication of your work at The EMBO Journal pending the above points in a time frame of two weeks.

Once we have received the revised version, we should then be able to swiftly complete formal acceptance and expedited production of the manuscript.

We also need you to take care of a number of minor issues related to formatting and data annotation, which I will share shortly in a separate message, together with additional changes and requests by our production team and my colleague H. Sonntag (CC'ed) for Source Data provision.

Please submit a revised version of the manuscript using the link enclosed below, addressing the advisor's comments.

As you might have seen on our web page, every paper at the EMBO Journal now includes a 'Synopsis', displayed on the html and freely accessible to all readers. The synopsis includes a 'model' figure as well as 2-5 one-short-sentence bullet points that summarize the article. I would appreciate if you could provide this figure and the bullet points.

Thank you again for giving us the chance to consider your manuscript for The EMBO Journal, I look forward to hearing from you and receiving your final revised version of the manuscript.

Kind regards,

Daniel Klimmeck

Daniel Klimmeck PhD
Senior Editor
The EMBO Journal.

Please use the link below to submit your revision:

EMBOJ-2024-117941, Arbitrating advisor's comments:

In my opinion this is an interesting, important and generally well executed study. Particularly the observations that p21 inhibition can alleviate lung fibrosis makes this a very important study. I believe that it is appropriate for EMBO J to further pursue this. In the attached document, I have included my comments to the reviewers second round of concerns and criticisms.

The Second Revision -

Point-by-point response to the Reviewers' comments following the first revision

Papismadov et al., "p21 connects senescence mechanisms with the ECM expression and promotes fibrosis"

Reviewer #1:

Remarks to the Author:

Some of the major concerns remain unaddressed. One of these is best illustrated with a statement in the first half of Abstract of the paper here highlighted:

"We show that CDK inhibitor p21 (CDKN1A), known to regulate the cell cycle and the viability of senescent cells, controls the expression of ECM components. The regulation of ECM component expression in both proliferating and senescent cells by p21 is CDK4- and Rb-dependent."

This has previously been demonstrated and reported (Sturmlechner et al. 2021) and as such cannot be claimed as novel. Specifically, please see Figures 1 and 2 and Tables 5, 6, and 8 of this paper. The unfounded claims of novelty were clearly stated as one of the major concerns on the manuscript, but these were not properly addressed by acknowledging and incorporating

the earlier findings.

AUTHORS' response: We agree that the findings in the Sturmlechner et al. Science, 2021 (which is extensively cited in our manuscript) are in complete agreement with our findings and, therefore, provide independent confirmation to the validity of our results and conclusions. In fact, we were very first to report that p21 regulates senescent cell viability (Yosef et al, EMBO J., 2017). Our current study focuses more on the regulation of the ECM component expression, its mechanism, and the in vivo effect. The study by Sturmlechner et al. mainly focuses on SASP regulation and even though changes in ECM components expression can be seen in the data, as underscored by the reviewer, it was not the focus of the study and was not discussed neither in the results nor in the discussion of this paper. In fact, the term ECM is mentioned only once throughout the paper by Sturmlechner et al. We would be happy to dedicate a sentence in the introduction to the above discussion in order to clarify the contributions as suggested by the reviewer and as it reflected in the published manuscripts. We will also go over our introduction and discussion to make sure that the paper by Sturmlechner et al. is cited correctly.

@Arbitrating advisor's comment: Data presented in this manuscript are sufficiently novel and distinct from the Sturmlechner 2021 paper. I agree with the authors that a mention of the Sturmlechner data (which only peripherally mentioned ECM production) in the introduction and discussion is adequate to resolve this concern.

The second remaining concern is the main emphasis of the connection between senescence and fibrosis as illustrated by the title ("p21 connects senescence mechanisms with ECM components expression and promotes fibrosis") and a statement in the Abstract ("Fibrosis and accumulation of senescent cells are common tissue changes associated with aging. Here we identified a molecular mechanism that connects the two phenomena"). As was pointed out, ECM components are expressed as an immediate early response to p21 induction (within 2 days) and are not dependent on cellular senescence. One would expect induction of ECM component expression after BLM treatment to follow p21 induction, which according to the authors can already be observed at day 3 (supplementary figure 10). Therefore, the authors' statement in the abstract "Fibrosis and accumulation of senescent cells are common tissue

changes associated with aging. Here we identified a molecular mechanism that connects the two phenomena" is premature because p21-mediated induction of ECM components is senescence independent (Sturmlechner et al., 2021). The authors included a day-10 post BLM treatment timepoint demonstrating that ECM components are already elevated at that stage. Unfortunately, day 10 is a timeframe that is more than sufficient for senescent cell transition, and day 3-4 timepoint would be needed to demonstrate that in the BLM model, ECM component expression is p21 dependent. The authors should include a day 3 or 4 (pre-senescent; immediate-early) post-BLM treatment timepoint in their analysis.

AUTHORS' response: We completely agree with the reviewer that p21 can regulate ECM expression independent of senescence. Our data in Fig3 and the corresponding suppl figure shows exactly this - ECM regulation by p21 in growing cells. In addition, the same figure shows that this effect is dependent on CDK4, similarly to the effect of CDK4 in senescent cells. Therefore, the same mechanism regulates ECM expression in senescent and growing cells. The exact same mechanism regulates cell cycle arrest in senescent cells, and the cells try to enter cell cycle upon p21 knockdown (Yosef et al, EMBO J. 2017). Therefore, our statement that this mechanism connects cell cycle arrest in senescent cells and regulation of ECM in all cells is based on the above data. The reviewer suggests performing Bleomycin induced lung fibrosis experiments at one addition time point- day 3 to possibly better resolve this point in vivo. However, according to his own explanation and our as well as his tissue culture data the expected result is - a difference in ECM at this point as is other points. Moreover, the ECM components expression is very very low at this point and thus testing these very low levels will not provide any new insights. In conclusion we are in full agreement with the reviewer that p21 can control ECM expression independent of senescence, it is clearly stated in the discussion already and we will make sure to emphasize this point further.

@Arbitrating advisor's comment: I agree with the reviewer that the statement in the abstract is not completely accurate. I suggest that authors change the sentence from "one mechanism connects cell cycle progression and the ECM components expression" to "one mechanism regulates cell cycle progression and expression of ECM components"

The remaining concerns were appropriately addressed. If the unaddressed concerns were

properly addressed, this reviewer would be supportive of publication.

AUTHORS' response: We are happy to see that this reviewer is supportive of the publication

Reviewer #2:

Remarks to the Author:

Papismanov et al present a revised version of their study "p21 connects senescence mechanisms with the ECM expression and promotes fibrosis" in which they demonstrate that p21 plays a critical role in the development of BLM-induced lung fibrosis. Despite some additional discussions and addition of some (minor) experiments, the revision does not really address most of the points raised by this reviewer during the first round.

AUTHORS' response: We are disappointed to see the negative approach of this review, and especially that this reviewer finds addition of substantial amount of in vivo data, including some of the analysis requested by this reviewer, as addition of minor experiments. This is simply a nasty remark.

1) Measurement of other senescence markers in p21+ cells remain extremely limited. The only addition is p16 staining, an assay often limited by the lack of specificity of antibodies used to detect it.

AUTHORS' response: We respectfully disagree with the statement that the measurement of senescence markers is limited, especially when it comes without any specific suggestions which markers the reviewer wants to see. We have tested p16 and p15 in addition to p21 itself and performed double staining for p16 and p21. We also show that such cells, stained for these additional markers, are not present in p21 knockouts. Moreover, p21 by itself is considered marker of senescence and several additional markers were analyzed by WB in the lung tissue of the mice and their expression was reduced in p21 knockout mice. The statement regarding lack of specificity of p16 antibodies is general and does not suggest

any solution as well. While the concern existed in the field for a long time, recent studies by several laboratories, including ours (Rachmian et al, Nature Neuroscience, 2024; Doolittle et al, Nature Comm, 2023) have successfully used knockout verified antibodies to detect p16, including on a single cell level. We used such verified antibodies, and therefore, stand behind our findings.

@Arbitrating advisor's comment: The authors have convincingly demonstrated that BLM-causes p21 mediated senescence and that BLM-induced senescence is diminished in p21 depleted cells. No concerns from me.

2) The fate of p21^{-/-} cells is not addressed at all. The authors should come up with other assays and models (in vivo/ex vivo -- it is not necessary to develop an entirely new tracking model), to be able to address this point. Analysis of existing RNAseq datasets and staining of already available tissues for a couple of apoptotic markers is definitely not sufficient to make any conclusion.

AUTHORS' response: We had not made any conclusions about the fate of knockout cells in vivo. The fate of the cells in the knockout mice might be an interesting follow-up project. However, this question is not discussed in our study, and it is not immediately connected to the main point of the study. Moreover, this is an unfair request at this time of the revision process, taking the amount of data that is already present in the manuscript. The requested study requires long and extensive experiments, including establishment of new experimental systems. Such studies are clearly beyond what the current study is about.

@Arbitrating advisor's comment: I agree with the authors that characterizing the fate of p21^{-/-} cells is beyond the scope of this study.

3) Measurements of lung physiological functions should include various functional parameters, not inflammatory staining and mouse weight

AUTHORS' response: Measuring physiological function of the lungs might be interesting, however it requires very special equipment and expertise, and therefore is not performed

routinely. These studies have to be performed on live mice. Specifically for our study this means that performing such measurements would require to repeat all the in vivo experiments again, on more than 100 of mice altogether, which is not practical.

@Arbitrating advisor's comment: Also here I agree with the authors that this is beyond the scope of this study

4) The mechanism by which p21 regulated ECM proteins remains very confusing. The only experiment added here is a validation of the knock-down efficiency (which was supposed to be there from the beginning)

AUTHORS' response: This is an interesting comment. The mechanism quite extensively studied in the paper. The results are presented in figures 3,4 and the corresponding supplementary figures. However, like in many other studies some questions remain to be answered in the future. We discussed these questions in the discussion section and would be happy to emphasize more the limitation of our findings and future perspective on this point. Of note, based on the reviewer 1 comments, reviewer 1 would say that the mechanism was already studied in Sturmlechner et al. Indeed, we rely in part on findings in this paper to discuss the mechanism we describe. Of interest might be similarities and differences in regulation of SASP and ECM by these mechanisms.

@Arbitrating advisor's comment: In my opinion, the mechanism has been characterized sufficiently. The main point is that p21 regulates ECM expression (through CDK4/Rb/SMAD) and senescence, which together creates excessive EMC and SASP, thereby promoting fibrosis. This has been demonstrated convincingly in this study.

5) It is surprising that the reply to the comment about lack of validation of BLM in cell culture was that both BLM and etoposide are DNA damaging agents that can drive DNA-damage induced cellular senescence. As the authors are likely aware, BLM and etoposide induce different types of DNA damage. Considering the variability of senescence-associated phenotypes originating from different DNA damage inducers, it is key to evaluate the right models.

What phenotype is due to the downregulation of p21, which is possibly achieved in both senescent vs non-senescent cells, and what is dependent on the elimination (or prevention) of senescence was not addressed, simply argued somehow superficially.

AUTHORS' response: The reviewer simply asks to repeat tissue culture experiments performed by induction of senescence in normal fibroblasts by Etoposide with the same induction but by Bleomycin. There is nothing about validation in these requested experiments. We used a classical model for senescence, which is used in tens if not hundreds of papers, to study the molecular mechanisms induced following p21 knockdown. We used Bleomycin model of lung fibrosis as a classical model which is well described in the literature and is well traceable. In our earlier study (Yosef et al, 2017) we have shown that the phenotype is dependent on DNA damage response. In our current study we show that the effect is mediated by CDK4 and Rb. The effect is in complete agreement with the results in similar, but not identical, experimental systems published in Sturmlechner et al. and stressed by Reviewer 1. Therefore, tissue culture experiments with Bleomycin are not expected to lead to results any different from what have been already observed with other stimuli in several experimental systems in different laboratories. Therefore, it is hard to assume that the specific DNA damaging drug would have a different effect, even if this drug induces DNA damage by a slightly different mechanism.

@Arbitrating advisor's comment: It has been extensively demonstrated that both bleo and eto create double stranded breaks that activate p21 expression and senescence. No data exist, to the best of my knowledge, suggesting that bleo and eto activate fundamentally different types of senescence responses. I do not share reviewers' 3 concerns.

Reviewer #3:

Remarks to the Author:

Our understanding of the processes by which senescence leads to progressive organ fibrosis remains incomplete. This paper from the Krizhanovsky lab reports their data linking p21 to ECM component expression and fibrosis, information of interest to researchers and clinicians.

Furthermore, they report that CDKN1A itself regulates the expression of ECM components via interactions with CDK4, with targeting of this mechanism reducing fibrosis deposition in models of lung fibrosis.

This manuscript has already undergone one round of peer review and been revised in response to the reviewer's comments. In my view the in vivo experiments are well constructed and presented in their current form, and I have confined my comments to the in vitro work.

Positives:

The group provides additional confirmation of the key role of senescence in initiating inflammation and fibrosis in the BLM model of lung fibrosis in vivo. I am aware that this manuscript has already undergone one round of peer review and been revised in response to the reviewers' comments, and in my view the in vivo experiments are well constructed and presented in their current form and I have confined my comments to the in vitro work. I have no concerns with the statistical tests used.

AUTHORS' response: We are glad to see that the reviewer, in agreement with reviewer 1, sees our in vivo work as well constructed and suitable for publication.

Limitations:

The importance of senescence in this context in vivo has already been documented - albeit in less detail and with less marked protective changes than those shown here - in keeping with the author's proposal that p21 itself is directly linked to these changes. The in vitro studies exploring the mechanisms underlying this are of key importance.

AUTHORS' response: While the role of senescence in lung fibrosis was already reported, our study goes way beyond this and shows that knockdown of p21 completely abolishes fibrosis development - a phenotype that goes way beyond what was reported for senescent cell elimination. We also describe mechanisms of regulation of ECM expression by p21, which are independent of senescence induction (as underscored by reviewer 1 and in agreement with a previous study). Therefore, the suggested mechanism is confirmed by at least two

independent studies. While further studies of the mechanism in vivo might be interesting, they are definitely beyond the scope of the current study. We will further clarify the point of the mechanisms and possible limitations of its interpretation in the Discussion.

Negatives:

A key stated advance in this paper is the connection claimed to have been made between senescence mechanisms and ECM components, specifically p21, CDK4 and Rb. If robust, this would be another more significant advance. Much of this arises from studies comparing irradiated fibroblasts with increased p21, and irradiated fibroblasts with increased p21, treated with sip21 inhibition - showing reduced levels of multiple ECM components in this setting. AUTHORS' response: In fact, the experimental system was classical senescence induction in culture by Etoposide treatment.

Of importance, sip21 treatment resulted in >50% cell death (which would be expected to be principally in the p21 expressing senescent cells, whilst sparing the non-senescent cells). When irradiated fibroblasts were treated with both p21 and CDK4 inhibition, cell death was not seen, and ECM expression returned to the same levels as p21^{hi} CDK4 intact cells. As irradiated fibroblasts with inhibited p21 and CDK4 have equivalent levels of ECM deposition to irradiated fibroblasts with intact, elevated p21 and CDK4 - I am unclear how the proposed mechanism has been correctly tested or proven. The same comments apply to Rb - where siRb treatment resulted in increased cell death. Therefore, it cannot be inferred with confidence that siRb has altered the phenotype of surviving senescent cells, as opposed to causing selective cell death and in doing so skewing the results towards the surviving, less activated fibroblasts.

Can the authors propose additional experiments, alterations to the abstract, results and conclusions to resolve these concerns?

AUTHORS' response: We have shown that ECM regulation by p21 is independent of senescence (in agreement with reviewer 1). We have also showed that Rb dependence of the effect occurs in both senescent and growing cells. Similar conclusions could be reached based on the data published at Sturmlechner et al. and stressed by reviewer 1. Therefore, our

conclusions regarding the role of Rb and CDK4 are confirmed by two independent sets of completely different experiments performed at different laboratories. Therefore, there is no concern that senescent cells death following p21 knockdown affects the overall conclusion regarding the mechanism. We will further clarify this in the discussion as already requested by reviewer 1.

@Arbitrating advisor's comment: As the authors also point out, the key experiments that control for the potential confounding effects of cell death, is that ECM production is reduced in proliferating cells that do not undergo cell death following p21 inhibition and restored upon simultaneous cdk4 inhibition. However, cell death nonetheless occurs in senescent cells, thus data should be discussed with this limitation in mind. A mention of this could be incorporated in the discussion.

Additional questions

Figure 3. The authors propose that p21 is essential for the survival of senescent cells.... and that its interaction with CKD4 is essential in mediating its ECM suppressing actions.. My concern with this data is their statement that knocking down p21 alone selectively killed senescent cells - so the data shown here is likely to be enriched for non-senescent, non-activated and non-ECM secreting fibroblasts. In contrast, p21 and CDK4 dual knockdown abolished the cell killing properties of isolated p21 knockdown, and irradiated, senescent fibroblasts now with dual p21 and cdk4 knockdown had equivalent fibrosis to the untreated senescent controls with raised p21, so these results would suggest that the p21<>CDK4 interaction, and indeed p21 itself are not contributing to fibrosis here (apart from p21's actions on cell survival).

In this context - how can the authors be sure that p21 is antifibrotic via cdk4 binding? I'd argue that they have shown that the p21-cdk4 binding is essential to the survival of fibroblasts after senescence induction, but not to fibrosis induction.

AUTHORS' response: The reviewer's question suggest that we need to better explain the results. However, it is evident from Figs 3,4 and the corresponding supp figures that the regulation of ECM components expression by CDK4 but not Rb occurs in both senescent and

growing cells. In growing cells there is no cell death following the knockdown of p21 and therefore such cell death cannot account for the observed changes in the ECM components expression. Moreover, in these experiments, on both RNA and protein level, the normalization was to expression of the control genes, which provide an internal control. Therefore, it does not seem possible that cell death affected overall results. Again, similar conclusions could be reached based on the data published at Sturmlechner et al. and stressed by reviewer 1.

@Arbitrating advisor's comment: As above, I agree with the authors here. A discussion of cell death as confounding factor would be useful, however.

Figure 4. Again - the study in 4b looks at the impact of p21 knockdown and of Rb knockdown in expression of ECM components. My concern with the p21 knockdown is as before - if these have selectively deleted the profibrotic cells as claimed- doesn't this confound the analysis and interpretation of data now based on the cells likely to be expressing less or no p21? The authors state that siRb also caused cell death, then go on to note that both siRb and the triple siRb, siCDK4 and sip21 knockdown is anti-fibrotic.

Can the authors determine whether the effects of sip21 and siRb are due promoting senescent cell death or alteration of the transcriptional behavior of the surviving cells?

AUTHORS' response: The comments to figures 3 and 4 relate to exactly the same point if cell death of senescent cells following the knockdown affected the results. It seems that the reviewer is missing the point that the same effect is observed in growing and senescent cells, thus removing the concern that the effect is due to senescent cells death. Once again, similar conclusions could be reached based on the data obtained by very different unrelated experimental systems published at Sturmlechner et al. The comments suggest the this point requires further clarification of our statements. We will definitely clarify this point more, even though it is already stated at the Discussion.

@Arbitrating advisor's comment: As authors acknowledge, a clarification would be useful.

Dear Dr Krizhanovsky,

This is to share that we have in the meantime still received input from a second arbitrating expert, who was not anticipated at this point to provide comments.

We are pleased to note that this expert, in line with the first advisor, endorses publication of your study at the EMBO Journal. Please note that while well taken, we decided to editorially overrule his/her optional point on comparing senescent cell depletion to p21 inducible invalidation.

We do agree that highlighting the advance of the current study better is advisable.

Please let me know if you have any questions related.

Best regards,
Daniel Klimmeck

Daniel Klimmeck PhD
Senior Editor
The EMBO Journal.

EMBOJ-2024-117941, Arbitrating advisor #2's comments:

This study provides compelling evidence for p21's critical role in lung fibrosis development, using both knockout and inducible knockdown mouse models. The main discovery is that p21 inactivation, even after fibrosis initiation, can ameliorate the fibrotic phenotype, reduce inflammation, and decrease senescence markers. This suggests p21 as a potential therapeutic target for pulmonary fibrosis. The work is consistent with the 2021 Science paper by Sturmlechner et al., which showed p21-mediated activation of ECM components through RB-dependent transcription. The current study from Krizhanovsky lab advances the field by:

- 1- Demonstrating p21's role in vivo in a disease-relevant model
- 2- Showing that p21 inactivation can reverse established fibrosis
- 3- Linking p21 to both senescence and ECM regulation in the context of fibrosis

The evaluation process was thorough. Although sometimes rather dry, reviewers raised valid concerns about mechanistic details and experimental design. The authors have addressed many of these issues through additional experiments and analysis. I believe this manuscript is suitable for publication in EMBO Journal, particularly if the authors:

- More explicitly highlight the novelty of their findings compared to the 2021 study, emphasizing the in vivo disease relevance and potential therapeutic implications
- Include experiments comparing senescent cell depletion to p21 inducible invalidation, as suggested by reviewers

These additions would strengthen the manuscript's impact and clarify its contribution to the field. In conclusion, this study provides valuable insights into the role of p21 in fibrosis and senescence, with important implications for understanding and potentiality.

Dear Dr Krizhanovsky,

Further to below message I am here enclosing the mentioned additional formatting adjustments required for your final resubmission.

Please let us know anytime should you have additional questions related.

Best regards,

Daniel Klimmeck

Daniel Klimmeck PhD
Senior Editor
The EMBO Journal.

>> Please limit keywords to your study to maximally five.

>> Provide a completed author checklist for your study.

>> Author Contributions: Please remove the author contributions information from the manuscript text. Note that CRediT has replaced the traditional author contributions section as of now because it offers a systematic machine-readable author contributions format that allows for more effective research assessment. and use the free text boxes beneath each contributing author's name to add specific details on the author's contribution.

More information is available in our guide to authors.
<https://www.embopress.org/page/journal/14602075/authorguide>

>> Adjust the title of the 'Declaration of Interests' section to 'Disclosure and Competing Interests Statement'.

>> Section order should be corrected as follows: title page with complete author information, abstract, keywords, introduction, results, discussion, methods, data availability section, acknowledgements, disclosure and competing interests statement, references, main figure legends, tables, expanded figure legends.'

>> Figures in separate files: Figures should be uploaded as individual, high resolution figure files.

>> Figure legends: recheck figure legends to match the figures for each of the panels.

>> Callouts: please correct the following mismatches: Fig 2B is called out before Fig 1B; there are callouts for Fig 1M-P but there are no such panels; Fig 2J is called out but does not exist; callout for Fig 2H is missing; callouts are there for Fig 3I and J but they don't exist; there are callouts for Fig 4I-L but they don't exist; callouts missing for Fig 5L; callouts are missing for Fig 7A-K.

>> Appendix file: Supplemental figures should be grouped with the corresponding legend underneath each figure, be renamed "Appendix Figure S1" etc., and a table of contents should be added to the PDF, with page numbers. The file should be labelled "Appendix".

>> Funding: information on funding is incomplete in our online system. All funders mentioned in the Acknowledgements section should also be entered into our system.

>> Reference format: needs correcting to alphabetical order and 10 authors listed before et al. .

>> Reagent Table: please convert Suppl. Tables S1-3 into a reagents table at the beginning of the Methods section.

>> Data availability section: please enter a Data availability section into the manuscript, merging the current Resource Availability information into one paragraph, stating 'No large-scale data amenable to data repository deposition were generated in this study.' .

>> Source data: please provide a completed source data checklist and source data files as to the separate instructions by my colleague Hannah Sonntag. Source data files need to be reorganized to one file/folder per figure and ZIPing for each main figure. For EV and/or appendix figures, ZIP together all source data.

>> Consider additional changes and comments from our production team as indicated below:

- Figure legends:

1. Please note that the figure 1k; 4f; 5c; 6b, j; does not contain a micrograph, kindly rectify the scale bar related information in the figure legend appropriately.
2. Please note that the figure 2h; 6h; does not contain a western blot, kindly rectify this information in the figure legend appropriately.
3. Please note that the figures 1m-o; 2i-j; 4i-l; is missing in the manuscript. This needs to be rectified.
4. Please note that the legend for figure 3i-j; 5l; is missing in the manuscript. This needs to be rectified.
5. Please note that the exact p values are not provided in the legends of figures 1d-e, g-l; 2a, c, e-f, h; 3d-g; 4b, d-f; 5d-e, g-j; 6a-k.
6. Please note that the scale bar needs to be defined for figures 2b, g; 5f.
7. Please note that in figure 1c; the scale bar unit should be corrected from μM to μm (in the figure legend).

Reply to arbitrating advisors' comments: (the current replies are in red)**EMBOJ-2024-117941, Arbitrating advisor's comments:**

In my opinion this is an interesting, important and generally well executed study. Particularly the observations that p21 inhibition can alleviate lung fibrosis makes this a very important study. I believe that it is appropriate for EMBO J to further pursue this. In the attached document, I have included my comments to the reviewers second round of concerns and criticisms.

The Second Revision -

Point-by-point response to the Reviewers' comments following the first revision

Papismadov et al., "p21 connects senescence mechanisms with the ECM expression and promotes fibrosis"

Reviewer #1:

Remarks to the Author:

Some of the major concerns remain unaddressed. One of these is best illustrated with a statement in the first half of Abstract of the paper here highlighted:

"We show that CDK inhibitor p21 (CDKN1A), known to regulate the cell cycle and the viability of senescent cells, controls the expression of ECM components. The regulation of ECM component expression in both proliferating and senescent cells by p21 is CDK4- and Rb-dependent."

This has previously been demonstrated and reported (Sturmlechner et al. 2021) and as such cannot be claimed as novel. Specifically, please see Figures 1 and 2 and Tables 5, 6, and 8 of this paper. The unfounded claims of novelty were clearly stated as one of the major concerns on the manuscript, but these were not properly addressed by acknowledging and incorporating the earlier findings.

AUTHORS' response: We agree that the findings in the Sturmlechner et al. Science,

2021 (which is extensively cited in our manuscript) are in complete agreement with our findings and, therefore, provide independent confirmation to the validity of our results and conclusions. In fact, we were very first to report that p21 regulates senescent cell viability (Yosef et al, EMBO J., 2017). Our current study focuses more on the regulation of the ECM component expression, its mechanism, and the in vivo effect. The study by Sturmlechner et al. mainly focuses on SASP regulation and even though changes in ECM components expression can be seen in the data, as underscored by the reviewer, it was not the focus of the study and was not discussed neither in the results nor in the discussion of this paper. In fact, the term ECM is mentioned only once throughout the paper by Sturmlechner et al. We would be happy to dedicate a sentence in the introduction to the above discussion in order to clarify the contributions as suggested by the reviewer and as it reflected in the published manuscripts. We will also go over our introduction and discussion to make sure that the paper by Sturmlechner et al. is cited correctly.

@Arbitrating advisor's comment: Data presented in this manuscript are sufficiently novel and distinct from the Sturmlechner 2021 paper. I agree with the authors that a mention of the Sturmlechner data (which only peripherally mentioned ECM production) in the introduction and discussion is adequate to resolve this concern.

Sturmlechner 2021 paper is now cited in the introduction, results and discussion sections.

The second remaining concern is the main emphasis of the connection between senescence and fibrosis as illustrated by the title ("p21 connects senescence mechanisms with ECM components expression and promotes fibrosis") and a statement in the Abstract ("Fibrosis and accumulation of senescent cells are common tissue changes associated with aging. Here we identified a molecular mechanism that connects the two phenomena"). As was pointed out, ECM components are expressed as an immediate early response to p21 induction (within 2 days) and are not dependent on cellular senescence. One would expect induction of ECM component expression after BLM treatment to follow p21 induction, which according to the authors can already be observed at day 3 (supplementary figure 10). Therefore, the authors' statement in the abstract "Fibrosis and accumulation of senescent cells are common tissue changes

associated with aging. Here we identified a molecular mechanism that connects the two phenomena" is premature because p21-mediated induction of ECM components is senescence independent (Sturmlechner et al., 2021). The authors included a day-10 post BLM treatment timepoint demonstrating that ECM components are already elevated at that stage. Unfortunately, day 10 is a timeframe that is more than sufficient for senescent cell transition, and day 3-4 timepoint would be needed to demonstrate that in the BLM model, ECM component expression is p21 dependent. The authors should include a day 3 or 4 (pre-senescent; immediate-early) post-BLM treatment timepoint in their analysis.

AUTHORS' response: We completely agree with the reviewer that p21 can regulate ECM expression independent of senescence. Our data in Fig3 and the corresponding suppl figure shows exactly this - ECM regulation by p21 in growing cells. In addition, the same figure shows that this effect is dependent on CDK4, similarly to the effect of CDK4 in senescent cells. Therefore, the same mechanism regulates ECM expression in senescent and growing cells. The exact same mechanism regulates cell cycle arrest in senescent cells, and the cells try to enter cell cycle upon p21 knockdown (Yosef et al, EMBO J. 2017). Therefore, our statement that this mechanism connects cell cycle arrest in senescent cells and regulation of ECM in all cells is based on the above data. The reviewer suggests performing Bleomycin induced lung fibrosis experiments at one addition time point- day 3 to possibly better resolve this point in vivo. However, according to his own explanation and our as well as his tissue culture data the expected result is - a difference in ECM at this point as is other points. Moreover, the ECM components expression is very very low at this point and thus testing these very low levels will not provide any new insights. In conclusion we are in full agreement with the reviewer that p21 can control ECM expression independent of senescence, it is clearly stated in the discussion already and we will make sure to emphasize this point further.

@Arbitrating advisor's comment: I agree with the reviewer that the statement in the abstract is not completely accurate. I suggest that authors change the sentence from "one mechanism connects cell cycle progression and the ECM components expression" to "one mechanism regulates cell cycle progression and expression of ECM components"

We changed the sentence in the abstract in accordance with the reviewer's request.

The remaining concerns were appropriately addressed. If the unaddressed concerns were properly addressed, this reviewer would be supportive of publication.

AUTHORS' response: We are happy to see that this reviewer is supportive of the publication

Reviewer #2:

Remarks to the Author:

Papismanov et al present a revised version of their study "p21 connects senescence mechanisms with the ECM expression and promotes fibrosis" in which they demonstrate that p21 plays a critical role in the development of BLM-induced lung fibrosis. Despite some additional discussions and addition of some (minor) experiments, the revision does not really address most of the points raised by this reviewer during the first round.

AUTHORS' response: We are disappointed to see the negative approach of this review, and especially that this reviewer finds addition of substantial amount of in vivo data, including some of the analysis requested by this reviewer, as addition of minor experiments. This is simply a nasty remark.

1) Measurement of other senescence markers in p21+ cells remain extremely limited. The only addition is p16 staining, an assay often limited by the lack of specificity of antibodies used to detect it.

AUTHORS' response: We respectfully disagree with the statement that the measurement of senescence markers is limited, especially when it comes without any specific suggestions which markers the reviewer wants to see. We have tested p16 and p15 in addition to p21 itself and performed double staining for p16 and p21. We also show that such cells, stained for these additional markers, are not present in p21 knockouts. Moreover, p21 by itself is considered marker of senescence and several

additional markers were analyzed by WB in the lung tissue of the mice and their expression was reduced in p21 knockout mice.

The statement regarding lack of specificity of p16 antibodies is general and does not suggest any solution as well. While the concern existed in the field for a long time, recent studies by several laboratories, including ours (Rachmian et al, Nature Neuroscience, 2024; Doolittle et al, Nature Comm, 2023) have successfully used knockout verified antibodies to detect p16, including on a single cell level. We used such verified antibodies, and therefore, stand behind our findings.

@Arbitrating advisor's comment: The authors have convincingly demonstrated that BLM-causes p21 mediated senescence and that BLM-induced senescence is diminished in p21 depleted cells. No concerns from me.

We thank the Arbitrating advisor for supporting our point of view.

2) The fate of p21^{-/-} cells is not addressed at all. The authors should come up with other assays and models (in vivo/ex vivo -- it is not necessary to develop an entirely new tracking model), to be able to address this point. Analysis of existing RNAseq datasets and staining of already available tissues for a couple of apoptotic markers is definitely not sufficient to make any conclusion.

AUTHORS' response: We had not made any conclusions about the fate of knockout cells in vivo. The fate of the cells in the knockout mice might be an interesting follow-up project. However, this question is not discussed in our study, and it is not immediately connected to the main point of the study. Moreover, this is an unfair request at this time of the revision process, taking the amount of data that is already present in the manuscript. The requested study requires long and extensive experiments, including establishment of new experimental systems. Such studies are clearly beyond what the current study is about.

@Arbitrating advisor's comment: I agree with the authors that characterizing the fate of p21^{-/-} cells is beyond the scope of this study.

We thank the Arbitrating advisor for supporting our point of view.

3) Measurements of lung physiological functions should include various functional parameters, not inflammatory staining and mouse weight

AUTHORS' response: Measuring physiological function of the lungs might be interesting, however it requires very special equipment and expertise, and therefore is not performed routinely. These studies have to be performed on live mice. Specifically for our study this means that performing such measurements would require to repeat all the in vivo experiments again, on more than 100 of mice altogether, which is not practical.

@Arbitrating advisor's comment: Also here I agree with the authors that this is beyond the scope of this study

We thank the Arbitrating advisor for supporting our point of view.

4) The mechanism by which p21 regulated ECM proteins remains very confusing. The only experiment added here is a validation of the knock-down efficiency (which was supposed to be there from the beginning)

AUTHORS' response: This is an interesting comment. The mechanism quite extensively studied in the paper. The results are presented in figures 3,4 and the corresponding supplementary figures. However, like in many other studies some questions remain to be answered in the future. We discussed these questions in the discussion section and would be happy to emphasize more the limitation of our findings and future perspective on this point. Of note, based on the reviewer 1 comments, reviewer 1 would say that the mechanism was already studied in Sturmlechner et al. Indeed, we rely in part on findings in this paper to discuss the mechanism we describe. Of interest might be similarities and differences in regulation of SASP and ECM by these mechanisms.

@Arbitrating advisor's comment: In my opinion, the mechanism has been characterized sufficiently. The main point is that p21 regulates ECM expression (through CDK4/Rb/SMAD) and senescence, which together creates excessive EMC and SASP,

thereby promoting fibrosis. This has been demonstrated convincingly in this study.

We thank the Arbitrating advisor for supporting our point of view.

5) It is surprising that the reply to the comment about lack of validation of BLM in cell culture was that both BLM and etoposide are DNA damaging agents that can drive DNA-damage induced cellular senescence. As the authors are likely aware, BLM and etoposide induce different types of DNA damage. Considering the variability of senescence-associated phenotypes originating from different DNA damage inducers, it is key to evaluate the right models.

What phenotype is due to the downregulation of p21, which is possibly achieved in both senescent vs non-senescent cells, and what is dependent on the elimination (or prevention) of senescence was not addressed, simply argued somehow superficially.

AUTHORS' response: The reviewer simply asks to repeat tissue culture experiments performed by induction of senescence in normal fibroblasts by Etoposide with the same induction but by Bleomycin. There is nothing about validation in these requested experiments.

We used a classical model for senescence, which is used in tens if not hundreds of papers, to study the molecular mechanisms induced following p21 knockdown. We used Bleomycin model of lung fibrosis as a classical model which is well described in the literature and is well traceable. In our earlier study (Yosef et al, 2017) we have shown that the phenotype is dependent on DNA damage response. In our current study we show that the effect is mediated by CDK4 and Rb. The effect is in complete agreement with the results in similar, but not identical, experimental systems published in Sturmlechner et al. and stressed by Reviewer 1. Therefore, tissue culture experiments with Bleomycin are not expected to lead to results any different from what have been already observed with other stimuli in several experimental systems in different laboratories. Therefore, it is hard to assume that the specific DNA damaging drug would have a different effect, even if this drug induces DNA damage by a slightly different mechanism.

@Arbitrating advisor's comment: It has been extensively demonstrated that both bleo and eto create double stranded breaks that activate p21 expression and senescence. No data exist, to the best of my knowledge, suggesting that bleo and eto activate fundamentally different types of senescence responses. I do not share reviewers' 3 concerns.

We thank the Arbitrating advisor for supporting our point of view.

Reviewer #3:

Remarks to the Author:

Our understanding of the processes by which senescence leads to progressive organ fibrosis remains incomplete. This paper from the Krizhanovsky lab reports their data linking p21 to ECM component expression and fibrosis, information of interest to researchers and clinicians. Furthermore, they report that CDKN1A itself regulates the expression of ECM components via interactions with CDK4, with targeting of this mechanism reducing fibrosis deposition in models of lung fibrosis.

This manuscript has already undergone one round of peer review and been revised in response to the reviewer's comments. In my view the in vivo experiments are well constructed and presented in their current form, and I have confined my comments to the in vitro work.

Positives:

The group provides additional confirmation of the key role of senescence in initiating inflammation and fibrosis in the BLM model of lung fibrosis in vivo. I am aware that this manuscript has already undergone one round of peer review and been revised in response to the reviewers comments, and in my view the in vivo experiments are well constructed and presented in their current form and I have confined my comments to the in vitro work. I have no concerns with the statistical tests used.

AUTHORS' response: We are glad to see that the reviewer, in agreement with reviewer

1, sees our in vivo work as well constructed and suitable for publication.

Limitations:

The importance of senescence in this context in vivo has already been documented - albeit in less detail and with less marked protective changes than those shown here - in keeping with the author's proposal that p21 itself is directly linked to these changes. The in vitro studies exploring the mechanisms underlying this are of key importance.

AUTHORS' response: While the role of senescence in lung fibrosis was already reported, our study goes way beyond this and shows that knockdown of p21 completely abolishes fibrosis development - a phenotype that goes way beyond what was reported for senescent cell elimination. We also describe mechanisms of regulation of ECM expression by p21, which are independent of senescence induction (as underscored by reviewer 1 and in agreement with a previous study). Therefore, the suggested mechanism is confirmed by at least two independent studies. While further studies of the mechanism in vivo might be interesting, they are definitely beyond the scope of the current study. We will further clarify the point of the mechanisms and possible limitations of its interpretation in the Discussion.

Negatives:

A key stated advance in this paper is the connection claimed to have been made between senescence mechanisms and ECM components, specifically p21, CDK4 and Rb. If robust, this would be another more significant advance. Much of this arises from studies comparing irradiated fibroblasts with increased p21, and irradiated fibroblasts with increased p21, treated with sip21 inhibition - showing reduced levels of multiple ECM components in this setting.

AUTHORS' response: In fact, the experimental system was classical senescence induction in culture by Etoposide treatment.

Of importance, sip21 treatment resulted in >50% cell death (which would be expected to be principally in the p21 expressing senescent cells, whilst sparing the non-senescent cells). When irradiated fibroblasts were treated with both p21 and CDK4 inhibition, cell

death was not seen, and ECM expression returned to the same levels as p21hi CDK4 intact cells. As irradiated fibroblasts with inhibited p21 and CDK4 have equivalent levels of ECM deposition to irradiated fibroblasts with intact, elevated p21 and CDK4 - I am unclear how the proposed mechanism has been correctly tested or proven. The same comments apply to Rb - where siRb treatment resulted in increased cell death.

Therefore, it cannot be inferred with confidence that siRb has altered the phenotype of surviving senescent cells, as opposed to causing selective cell death and in doing so skewing the results towards the surviving, less activated fibroblasts.

Can the authors propose additional experiments, alterations to the abstract, results and conclusions to resolve these concerns?

AUTHORS' response: We have shown that ECM regulation by p21 is independent of senescence (in agreement with reviewer 1). We have also showed that Rb dependence of the effect occurs in both senescent and growing cells. Similar conclusions could be reached based on the data published at Sturmlechner et al. and stressed by reviewer 1. Therefore, our conclusions regarding the role of Rb and CDK4 are confirmed by two independent sets of completely different experiments performed at different laboratories. Therefore, there is no concern that senescent cells death following p21 knockdown affects the overall conclusion regarding the mechanism. We will further clarify this in the discussion as already requested by reviewer 1.

@Arbitrating advisor's comment: As the authors also point out, the key experiments that control for the potential confounding effects of cell death, is that ECM production is reduced in proliferating cells that do not undergo cell death following p21 inhibition and restored upon simultaneous cdk4 inhibition. However, cell death nonetheless occurs in senescent cells, thus data should be discussed with this limitation in mind. A mention of this could be incorporated in the discussion.

As requested, we added the following sentences to the discussion, to paragraph 3. The additional sentences are as follows:

“Knockdown of p21 in senescent cells leads to the induction of apoptosis. Consequently, the observed decrease in extracellular matrix (ECM) component production may be attributable to the loss of senescent cells. However, p21 also regulates ECM production

in proliferating cells, where apoptosis does not occur.”

Additional questions

Figure 3. The authors propose that p21 is essential for the survival of senescent cells.... and that its interaction with CKD4 is essential in mediating its ECM suppressing actions.. My concern with this data is their statement that knocking down p21 alone selectively killed senescent cells - so the data shown here is likely to be enriched for non-senescent, non-activated and non-ECM secreting fibroblasts. In contrast, p21 and CDK4 dual knockdown abolished the cell killing properties of isolated p21 knockdown, and irradiated, senescent fibroblasts now with dual p21 and cdk4 knockdown had equivalent fibrosis to the untreated senescent controls with raised p21, so these results would suggest that the p21<>CDK4 interaction, and indeed p21 itself are not contributing to fibrosis here (apart from p21's actions on cell survival).

In this context - how can the authors be sure that p21 is antifibrotic via cdk4 binding? I'd argue that they have shown that the p21-cdk4 binding is essential to the survival of fibroblasts after senescence induction, but not to fibrosis induction.

AUTHORS' response: The reviewer's question suggest that we need to better explain the results. However, it is evident from Figs 3,4 and the corresponding supp figures that the regulation of ECM components expression by CDK4 but not Rb occurs in both senescent and growing cells. In growing cells there is no cell death following the knockdown of p21 and therefore such cell death cannot account for the observed changes in the ECM components expression. Moreover, in these experiments, on both RNA and protein level, the normalization was to expression of the control genes, which provide an internal control. Therefore, it does not seem possible that cell death affected overall results. Again, similar conclusions could be reached based on the data published at Sturmlechner et al. and stressed by reviewer 1.

@Arbitrating advisor's comment: As above, I agree with the authors here. A discussion of cell death as confounding factor would be useful, however.

This issue was further explained in the discussion. Please see the comment above.

Figure 4. Again - the study in 4b looks at the impact of p21 knockdown and of Rb knockdown in expression of ECM components. My concern with the p21 knockdown is as before - if these have selectively deleted the profibrotic cells as claimed- doesn't this confound the analysis and interpretation of data now based on the cells likely to be expressing less or no p21? The authors state that siRb also caused cell death, then go on to note that both siRb and the triple siRb, siCDK4 and sip21 knockdown is anti-fibrotic.

Can the authors determine whether the effects of sip21 and siRb are due promoting senescent cell death or alteration of the transcriptional behavior of the surviving cells?

AUTHORS' response: The comments to figures 3 and 4 relate to exactly the same point if cell death of senescent cells following the knockdown affected the results. It seems that the reviewer is missing the point that the same effect is observed in growing and senescent cells, thus removing the concern that the effect is due to senescent cells death. Once again, similar conclusions could be reached based on the data obtained by very different unrelated experimental systems published at Sturmlechner et al. The comments suggest the this point requires further clarification of our statements. We will definitely clarify this point more, even though it is already stated at the Discussion.

@Arbitrating advisor's comment: As authors acknowledge, a clarification would be useful.

This issue was further explained in the discussion. Please see the comment above.

EMBOJ-2024-117941, Arbitrating advisor #2's comments:

This study provides compelling evidence for p21's critical role in lung fibrosis development, using both knockout and inducible knockdown mouse models. The main discovery is that p21 inactivation, even after fibrosis initiation, can ameliorate the fibrotic phenotype, reduce inflammation, and decrease senescence markers. This suggests p21 as a potential therapeutic target for pulmonary fibrosis. The work is consistent with the 2021 Science paper by Sturmlechner et al., which showed p21-mediated activation of

ECM components through RB-dependent transcription. The current study from Krizhanosky lab advances the field by:

- 1- Demonstrating p21's role in vivo in a disease-relevant model
- 2- Showing that p21 inactivation can reverse established fibrosis
- 3- Linking p21 to both senescence and ECM regulation in the context of fibrosis

The evaluation process was thorough. Although sometimes rather dry, reviewers raised valid concerns about mechanistic details and experimental design. The authors have addressed many of these issues through additional experiments and analysis. I believe this manuscript is suitable for publication in EMBO Journal, particularly if the authors:

We thank the Arbitrating advisor for supporting publication of our study at EMBO Journal.

- More explicitly highlight the novelty of their findings compared to the 2021 study, emphasizing the in vivo disease relevance and potential therapeutic implications
- Include experiments comparing senescent cell depletion to p21 inducible invalidation, as suggested by reviewers

These additions would strengthen the manuscript's impact and clarify its contribution to the field. In conclusion, this study provides valuable insights into the role of p21 in fibrosis and senescence, with important implications for understanding and potentiality.

To address the advisor comment and emphasize stronger the disease relevance we modified the abstract and added the following sentences to the discussion, to paragraph 6.

“Additionally, p21 is essential in modulating the secretome to affect the immunosurveillance of stressed cells (Sturmlechner *et al.*, 2021). Therefore, we suggest that p21 silencing may not only limit the damage induced by the presence of senescent cells due to their elimination (Yosef *et al.*, 2017), but also promote the recovery from tissue damage via modulation of the cellular microenvironment and reduction in the ECM production and inflammation in both proliferating and senescent cells. Overall, p21 promotes fibrosis and controls central molecular mechanisms regulating ECM

expression and the viability of senescent cells. Therefore, inhibition of p21 is a plausible strategy for efficient treatment of fibrotic and non-fibrotic age-related pathologies.”

EMBOJ-2024-117941, Papismadov et al.

Reply to Comments by the Editor (our replies are in red):

>> Please limit keywords to your study to maximally five.

We reduced the number of keywords from 6 to 5. The keyword that was removed is "Rb".

>> Provide a completed author checklist for your study.

A complete author checklist has been provided.

>> Author Contributions: Please remove the author contributions information from the manuscript text. Note that CRediT has replaced the traditional author contributions section as of now because it offers a systematic machine-readable author contributions format that allows for more effective research assessment. and use the free text boxes beneath each contributing author's name to add specific details on the author's contribution.

More information is available in our guide to authors.

The 'Author contributions' section was removed from the manuscript.

>> Adjust the title of the 'Declaration of Interests' section to 'Disclosure and Competing Interests Statement'.

The title was adjusted in the manuscript.

>> Section order should be corrected as follows: title page with complete author information, abstract, keywords, introduction, results, discussion, methods, data availability section, acknowledgements, disclosure and competing interests statement, references, main figure legends, tables, expanded figure legends.'

The order of all section was corrected and now follows EMBO Journal guidelines.

>> Figures in separate files: Figures should be uploaded as individual, high resolution

figure files.

All figures are available in either PDF or TIFF formats and will be uploaded individually.

>> Figure legends: recheck figure legends to match the figures for each of the panels.

All figure legends, both for the main figures and for the appendix figures have been rechecked.

>> Callouts: please correct the following mismatches: Fig 2B is called out before Fig 1B; there are callouts for Fig 1M-P but there are no such panels; Fig 2J is called out but does not exist; callout for Fig 2H is missing; callouts are there for Fig 3I and J but they don't exist; there are callouts for Fig 4I-L but they don't exist; callouts missing for Fig 5L; callouts are missing for Fig 7A-K.

All the callouts have been modified in this version of the manuscript.

>> Appendix file: Supplemental figures should be grouped with the corresponding legend underneath each figure, be renamed "Appendix Figure S1" etc., and a table of contents should be added to the PDF, with page numbers. The file should be labelled "Appendix".

All supplemental figures and their figure legends have been changed to Appendix figures and are placed in a folder named "Appendix".

>> Funding: information on funding is incomplete in our online system. All funders mentioned in the Acknowledgements section should also be entered into our system.

All funders mentioned in the Acknowledgements section would be entered.

>> Reference format: needs correcting to alphabetical order and 10 authors listed before et al. .

Reference format has been changed.

>> Reagent Table: please convert Suppl. Tables S1-3 into a reagents table at the beginning of the Methods section.

Done as requested. See in Materials and Methods section in the manuscript.

>> Data availability section: please enter a Data availability section into the manuscript,

merging the current Resource Availability information into one paragraph, stating 'No large-scale data amenable to data repository deposition were generated in this study.'

Data availability section was added to the manuscript.

>> Source data: please provide a completed source data checklist and source data files as to the separate instructions by my colleague Hannah Sonntag. Source data files need to be reorganized to one file/folder per figure and ZIPing for each main figure. For EV and/or appendix figures, ZIP together all source data.

A completed source data checklist filled and source data files have been created for each figure separately.

>> Consider additional changes and comments from our production team as indicated below:

- Figure legends:

1. Please note that the figure 1k; 4f; 5c; 6b, j; does not contain a micrograph, kindly rectify the scale bar related information in the figure legend appropriately.

Micrograph have been added to all figures.

The updated figures are as follows:

Figure 1c, 2b, 2g, 5f, 7b, 7j

2. Please note that the figure 2h; 6h; does not contain a western blot, kindly rectify this information in the figure legend appropriately.

In the revised figures, the legends contain the correct information. The updated figures are as follows:

Figure 3h (instead of 2h) and figure 7h (instead of 6h)

3. Please note that the figures 1m-o; 2i-j; 4i-l; is missing in the manuscript. This needs to be rectified.

In the revised figures, all figures are mentioned in the text and in the figure legends.

4. Please note that the legend for figure 3i-j; 5l; is missing in the manuscript. This needs to be rectified.

In the revised figures, the legends contain the correct information.

5. Please note that the exact p values are not provided in the legends of figures 1d-e, g-l; 2a, c, e-f, h; 3d-g; 4b, d-f; 5d-e, g-j; 6a-k.

P values have been added to all graphs mentioned by the Editor and also in the appendix figures.

6. Please note that the scale bar needs to be defined for figures 2b, g; 5f.

All scale bars have been updated, both in the main figures and in the Appendix figures.

7. Please note that in figure 1c; the scale bar unit should be corrected from μM to μm (in the figure legend).

All figure legends with this mistake have been corrected.

Dear Valery,

Thank you for submitting the revised version of your manuscript. I have now evaluated your amended manuscript and related files and concluded that the remaining concerns have been sufficiently addressed.

I am thus pleased to inform you that your manuscript has been accepted for publication in the EMBO Journal.

On a different note, I would like to alert you that EMBO Press offers a format for a video-synopsis of work published with us, which essentially is a short, author-generated film explaining the core findings in hand drawings, and, as we believe, can be very useful to increase visibility of the work. Please see the following link for representative examples and their integration into the article web page:

<https://www.embopress.org/doi/full/10.15252/emj.2019103932>

Best regards,

Daniel

Daniel Klimmeck, PhD
Senior Editor
The EMBO Journal
EMBO
Postfach 1022-40
Meyerhofstrasse 1
D-69117 Heidelberg
contact@embojournal.org
Submit at: <http://emboj.msubmit.net>